

# Yang-Baxter integrable Lindblad equations

**Aleksandra A. Ziolkowska and Fabian H.L. Essler**

The Rudolf Peierls Centre for Theoretical Physics,
Oxford University, Oxford OX1 3PU, UK

## Abstract

We consider Lindblad equations for one dimensional fermionic models and quantum spin chains. By employing a (graded) super-operator formalism we identify a number of Lindblad equations than can be mapped onto non-Hermitian interacting Yang-Baxter integrable models. Employing Bethe Ansatz techniques we show that the late-time dynamics of some of these models is diffusive.


doi:[10.21468/SciPostPhys.8.3.044]()

# 1   Introduction

Weak couplings to an environment can have very interesting effects on the dynamics of many-particle quantum systems. In particular they can result in desirable non-equilibrium steady states [1–5]. In order to arrive at a tractable theoretical description it is customary to employ a Markovian approximation that assumes that the characteristic times scales associated with the environment are much shorter than those of the many-particle system of interest. The absence of a back action of the system onto its environment then facilitates a well defined mathematical description of open many-particle systems. In the quantum case this a priori results in a Markovian quantum stochastic many-particle system [6–9], which is however difficult to analyze. The customary approach is therefore to focus on the dynamics averaged over the environment, which leads to a description by the Lindblad master equation [10] for the time-dependent reduced density matrix $\rho(t)$

$$\frac{d\rho}{dt} = i[\rho, H] + \sum_a \gamma_a \left[ L_a \rho L_a^\dagger - \frac{1}{2} \{L_a^\dagger L_a, \rho\} \right]. \tag{1}$$

Here $H$ is the system Hamiltonian, $L_a$ are jump operators that encode the coupling to the environment and $\gamma_a > 0$. While much progress has been made in analyzing Lindblad equations for many-particle systems by employing e.g. perturbative [11,12] and matrix product states methods [13–16] it clearly is highly desirable to have exact solutions in specific, and hopefully representative, cases. In the context of master equations for classical stochastic many

particle systems an example of such a solvable paradigm is the asymmetric simple exclusion process [17–22]. In the quantum case it has been known for some time that certain Lindblad equations describing many-particle systems can be represented by Liouvillians that are quadratic in fermionic or bosonic creation and annihilation operators, which makes it possible to solve them exactly by elementary means [23–26]. Very recently examples of Lindblad equations with Liouvillians related to *interacting* Yang-Baxter integrable models have been found [27–29]. This opens the door for bringing quantum integrability methods to bear on obtaining exact results for the dynamics of open many-particle quantum systems. An obvious question is whether the known cases are exceptional, or whether there are other examples of Yang-Baxter integrable Lindblad equations. In this work we report on the results of a search for integrable cases among a particular class of Lindblad equations for translationally invariant many-particle quantum systems.

## 2 Lindblad equations for lattice models

We now turn to the precise definition of the class of quantum master equations we will be interested in. We consider one dimensional lattice models with local Hilbert spaces that can include bosonic as well as fermionic degrees of freedom. A basis of the local Hilbert space is formed by $N$ bosonic and $M$ fermionic quantum states

$$|\alpha\rangle_j , \quad \alpha = 1, \dots, N + M . \tag{2}$$

We denote the fermion parity of the state $|\alpha\rangle_j$ by $\epsilon_\alpha$

$$\epsilon_\alpha = \begin{cases} 0 & \text{if } \alpha \text{ is bosonic} \\ 1 & \text{if } \alpha \text{ is fermionic} \end{cases} . \tag{3}$$

An orthonormal basis of the full Hilbert space $\mathcal{H}_L$ on an $L$-site chain is then given by the states

$$|\boldsymbol{\alpha}\rangle \equiv \otimes_{j=1}^{L} |\alpha_j\rangle_j , \quad \alpha_j \in \{1, \dots, N + M\} . \tag{4}$$

We define the fermion parity of the states (4) by

$$\epsilon_{\boldsymbol{\alpha}} = \sum_{j=1}^{L} \epsilon_{\alpha_j} . \tag{5}$$

A basis of the space of linear operators acting on site $j$ is then provided by

$$E_j^{\alpha\beta} = |\alpha\rangle_j \, {}_j\langle\beta| , \quad \alpha, \beta \in \{1, \dots, N + M\} . \tag{6}$$

These are often referred to as Hubbard operators. Their fermion parity is $\epsilon_\alpha + \epsilon_\beta$ mod 2, i.e. they are fermionic if either the state $|\alpha\rangle$ or the state $|\beta\rangle$ is fermionic. The operators $E_n^{\alpha\beta}$ act on the states $|\boldsymbol{\alpha}\rangle$ as

$$E_n^{\alpha\beta}|\boldsymbol{\alpha}\rangle = (-1)^{(\epsilon_\alpha + \epsilon_\beta)\sum_{j=1}^{n-1} \epsilon_{\alpha_j}} \delta_{\beta, \alpha_n} |\boldsymbol{\alpha}'\rangle , \quad \boldsymbol{\alpha}' = \alpha_1, \dots, \alpha_{n-1}, \alpha, \alpha_{n+1}, \dots, \alpha_L . \tag{7}$$

Minus signs are acquired when moving fermionic operators past fermionic states. The operators defined in this way either commute or anticommute on different sites

$$E_j^{\alpha\beta} E_k^{\gamma\delta} = (-1)^{(\epsilon_\alpha + \epsilon_\beta)(\epsilon_\gamma + \epsilon_\delta)} E_k^{\gamma\delta} E_j^{\alpha\beta} , \quad k \neq j . \tag{8}$$

For later convenience we define a *graded permutation operator* on sites $j$ and $j+1$

$$\Pi_{j,j+1} = \sum_{\alpha,\beta} (-1)^{\epsilon_\beta} E_j^{\alpha\beta} E_{j+1}^{\beta\alpha} \, . \tag{9}$$

It acts on states as

$$\Pi_{j,j+1}|\beta\rangle_j|\alpha\rangle_{j+1} \;=\; (-1)^{\epsilon_\alpha \epsilon_\beta}|\alpha\rangle_j|\beta\rangle_{j+1} \, , \tag{10}$$

i.e. it permutes the states and generates a minus sign if both states are fermionic.

## 2.1 A useful decomposition for $N = n_B^2 + n_F^2$, $M = 2n_B n_F$ for integer $n_B$, $n_F$

The general local Hilbert space $V_{N+M}$ introduced above has $N$ bosonic and $M$ fermionic basis states. If $N$ and $M$ are such that they can be expressed as $N = n_B^2 + n_F^2$ and $M = 2n_B n_F$ for integer $n_B$, $n_F$ it is possible to express $V_{N+M}$ as a graded tensor product of two $n = n_B + n_F$-dimensional spaces $V_{N+M} = V_n \otimes V_n$. Here $n_B$ and $n_F$ are the numbers of bosonic and fermionic basis states of $V_n$. Denoting the basis of $V_n$ by $\{|1\rangle, \ldots, |n\rangle\}$ we can express the $N + M$ basis states of $V_{N+M}$ as

$$|\alpha\rangle = |\widetilde{\alpha}\rangle \otimes |\bar{\alpha}\rangle \, , \quad \alpha = 1, \ldots, N+M \, , \tag{11}$$

where $1 \leq \bar{\alpha}, \widetilde{\alpha} \leq n$ are related to $\alpha$ by

$$\bar{\alpha} = \alpha \bmod n + n\delta_{\alpha \bmod n, 0} \, , \qquad \widetilde{\alpha} = \left\lfloor \frac{\alpha}{n+1} \right\rfloor + 1 \, . \tag{12}$$

We note that $\alpha = n(\widetilde{\alpha} - 1) + \bar{\alpha}$ and that the fermion parities are related by $\epsilon_\alpha = \epsilon_{\widetilde{\alpha}} + \epsilon_{\bar{\alpha}}$. Defining operators

$$\widetilde{e}_j^{\widetilde{\alpha}\widetilde{\beta}} = |\widetilde{\alpha}\rangle_j {}_j\langle\widetilde{\beta}| \, , \quad e_j^{\bar{\alpha}\bar{\beta}} = |\bar{\alpha}\rangle_j {}_j\langle\bar{\beta}| \, , \tag{13}$$

we may express $E_j^{\alpha\beta}$ in the form

$$E_j^{\alpha\beta} = |\alpha\rangle\langle\beta| = |\widetilde{\alpha}\rangle|\bar{\alpha}\rangle\langle\bar{\beta}|\langle\widetilde{\beta}| = (-1)^{\epsilon_{\widetilde{\beta}}(\epsilon_{\bar{\alpha}} + \epsilon_{\bar{\beta}})} \widetilde{e}_j^{\widetilde{\alpha}\widetilde{\beta}} \, e_j^{\bar{\alpha}\bar{\beta}} \, . \tag{14}$$

We will use this decomposition in several models considered below. In the purely bosonic case $M = 0$ such decompositions are possible for $N = n^2$ with integer $n$.

## 2.2 Super-operator formalism for Lindblad equations

We now consider a Lindblad equation (1) with a Hamiltonian $H$ and jump operators $L_a$ acting on $\mathcal{H}_L$ defined above. We are ultimately interested in cases where the Hamiltonian density and $L_a$ have local expansions in terms of the $E_j^{\alpha\beta}$. To start with we will assume for simplicity that all jump operators are bosonic. The cases where some of the jump operators are fermionic will be discussed later. The reduced density matrix can be expressed in terms of the basis states defined above as

$$\rho = \sum_{\boldsymbol{\alpha},\boldsymbol{\beta}} \rho_{\boldsymbol{\alpha},\boldsymbol{\beta}}|\boldsymbol{\alpha}\rangle\langle\boldsymbol{\beta}| \, . \tag{15}$$

The matrix elements are related to particular Green's functions of the operators $E_j^{\alpha\beta}$

$$\rho_{\boldsymbol{\alpha},\boldsymbol{\beta}} = (-1)^{\sum_{j=1}^{L-1} \sum_{k=j+1}^{L} \epsilon_{\beta_j}(\epsilon_{\beta_k} + \epsilon_{\alpha_k})} \text{Tr}\left[\rho \, E_L^{\beta_L \alpha_L} \ldots E_1^{\beta_1 \alpha_1}\right] \, . \tag{16}$$

In terms of components the Lindblad equation reads

$$\frac{d}{dt}\rho_{\boldsymbol{\alpha},\boldsymbol{\beta}} = i\sum_{\boldsymbol{\gamma}}\rho_{\boldsymbol{\alpha},\boldsymbol{\gamma}}H_{\boldsymbol{\gamma},\boldsymbol{\beta}} - H_{\boldsymbol{\alpha},\boldsymbol{\gamma}}\rho_{\boldsymbol{\gamma},\boldsymbol{\beta}}$$

$$+ \sum_{a}\gamma_a\Big\{\sum_{\boldsymbol{\gamma},\boldsymbol{\delta}}(L_a)_{\boldsymbol{\alpha},\boldsymbol{\gamma}}\rho_{\boldsymbol{\gamma},\boldsymbol{\delta}}(L_a^\dagger)_{\boldsymbol{\delta},\boldsymbol{\beta}} - \frac{1}{2}\sum_{\boldsymbol{\gamma}}(L_a^\dagger L_a)_{\boldsymbol{\alpha},\boldsymbol{\gamma}}\rho_{\boldsymbol{\gamma},\boldsymbol{\beta}} + \rho_{\boldsymbol{\alpha},\boldsymbol{\gamma}}(L_a^\dagger L_a)_{\boldsymbol{\gamma},\boldsymbol{\beta}}\Big\}, \tag{17}$$

where we have introduced the following notations for the matrix elements of an operator $\mathcal{O}$

$$\langle\boldsymbol{\alpha}|\mathcal{O}|\boldsymbol{\beta}\rangle = \mathcal{O}_{\boldsymbol{\alpha},\boldsymbol{\beta}}. \tag{18}$$

We can view the density matrix as a state in a $(N+M)^{2L}$ dimensional Hilbert space $\mathcal{H}_S = \mathcal{H}_L\otimes\mathcal{H}_L$ with basis states

$$|\boldsymbol{\alpha}\rangle|\boldsymbol{\beta}\rangle\!\rangle = |\alpha_1\rangle_1 \ldots|\alpha_L\rangle_L |\beta_1\rangle\!\rangle_1 \ldots|\beta_L\rangle\!\rangle_L. \tag{19}$$

In these notations we have

$$|\rho\rangle = \sum_{\boldsymbol{\alpha},\boldsymbol{\beta}}\rho_{\boldsymbol{\alpha},\boldsymbol{\beta}}|\boldsymbol{\alpha}\rangle|\boldsymbol{\beta}\rangle\!\rangle, \tag{20}$$

and the "wave-functions" $\rho_{\boldsymbol{\alpha},\boldsymbol{\beta}}$ correspond to Green's functions in the original problem. The Lindblad equation (17) can be cast in the form

$$\frac{d|\rho\rangle}{dt} = \mathcal{L}|\rho\rangle, \tag{21}$$

where the Liouvillian $\mathcal{L}$ for bosonic jump operators $L_a$ is given by

$$\mathcal{L} = -iH + i\bar{H} + \sum_a\gamma_a\left[L_a\overline{L_a^\dagger} - \frac{1}{2}\left(L_a^\dagger L_a + \overline{L_a^\dagger L_a}\right)\right]. \tag{22}$$

Here we employ notations such that $\mathcal{O} = \mathcal{O}\otimes\mathbb{1}$ and have defined related operators $\overline{\mathcal{O}} = \mathbb{1}\otimes\overline{\mathcal{O}}$ by

$$\langle\!\langle\boldsymbol{\gamma}|\overline{\mathcal{O}}|\boldsymbol{\beta}\rangle\!\rangle = \langle\boldsymbol{\beta}|\mathcal{O}|\boldsymbol{\gamma}\rangle. \tag{23}$$

One can easily check that taking the scalar product of (21) with the state $\langle\!\langle\boldsymbol{\beta}|\langle\boldsymbol{\alpha}|$ precisely reproduces (17). A convenient basis for expanding operators $\overline{\mathcal{O}}$ is constructed in terms of operators $\widetilde{E}_n^{\alpha\beta}$ defined as

$$\widetilde{E}_n^{\alpha\beta} = \mathbb{1}\otimes\big(|\alpha\rangle\!\rangle_n {}_n\langle\!\langle\beta|\big). \tag{24}$$

These act on basis states according to

$$\widetilde{E}_n^{\alpha\beta}|\boldsymbol{\alpha}\rangle|\boldsymbol{\beta}\rangle\!\rangle = (-1)^{(\epsilon_\alpha+\epsilon_\beta)\epsilon_{\boldsymbol{\alpha}}}|\boldsymbol{\alpha}\rangle\,\widetilde{E}_n^{\alpha\beta}|\boldsymbol{\beta}\rangle\!\rangle$$

$$= (-1)^{(\epsilon_\alpha+\epsilon_\beta)\epsilon_{\boldsymbol{\alpha}}}(-1)^{(\epsilon_\alpha+\epsilon_\beta)\sum_{j=1}^{n-1}\epsilon_{\beta_j}}\delta_{\beta,\beta_n}|\boldsymbol{\alpha}\rangle\|\boldsymbol{\beta}'\rangle\!\rangle, \tag{25}$$

where $|\boldsymbol{\beta}'\rangle\!\rangle = |\beta_1\rangle_1,\ldots,|\alpha\rangle_n,\ldots,|\beta_L\rangle_L$ and $\epsilon_{\boldsymbol{\alpha}}$ has been defined in (5). We note that the operators $E_n^{\alpha\beta}$ act on $\mathcal{H}_S$ as $E_n^{\alpha\beta}\otimes\mathbb{1}$.

## 2.3 Fermionic jump operators

If some of the jump operators are fermionic the super-operator formalism needs to be modified. Let us denote the fermion parity of the jump operator $L_a$ by $\epsilon_{L_a}\in\{0,1\}$. When written in

components the Lindblad equation still takes the form (17). However, the Liouvillian (22) is now replaced by

$$\mathcal{L} = -iH + i\bar{H} + \sum_a \gamma_a \left[ (-i)^{\epsilon_{L_a}} L_a \overline{L_a^\dagger} - \frac{1}{2}\left( L_a^\dagger L_a + \overline{L_a^\dagger L_a} \right) \right] . \tag{26}$$

The state representing the density matrix is also modified and now takes the form

$$|\rho\rangle = \sum_{\boldsymbol{\alpha},\boldsymbol{\beta}} \rho_{\boldsymbol{\alpha},\boldsymbol{\beta}} \left[ (-i)^{\epsilon_{\boldsymbol{\alpha}}} P_+ + i^{\epsilon_{\boldsymbol{\beta}}} P_- \right] |\boldsymbol{\alpha}\rangle|\boldsymbol{\beta}\rangle\!\rangle , \tag{27}$$

where $P_\pm$ are projection operators onto states with even and odd fermion parity respectively

$$P_\pm = \frac{1 \pm (-1)^F}{2} , \quad (-1)^F = \prod_{\ell=1}^{L} \prod_{\substack{\alpha=1 \\ \epsilon_\alpha=1}}^{N+M} (1 - 2E_\ell^{\alpha\alpha})(1 - 2\widetilde{E}_\ell^{\alpha\alpha}) . \tag{28}$$

We have

$$(-1)^F |\boldsymbol{\alpha}\rangle|\boldsymbol{\beta}\rangle\!\rangle = (-1)^{\epsilon_{\boldsymbol{\alpha}}+\epsilon_{\boldsymbol{\beta}}} |\boldsymbol{\alpha}\rangle|\boldsymbol{\beta}\rangle\!\rangle . \tag{29}$$

It is straightforward to check that inserting (26) and (27) into the equation

$$\frac{d}{dt}|\rho\rangle = \mathcal{L}|\rho\rangle , \tag{30}$$

and expanding it in a basis of states precisely recovers (17). We stress that in our construction both bosonic and fermionic jump operators can be accommodated as long as any given jump operator has a definite fermion parity.

# 3 Lindblad equations as non-Hermitian two-leg ladders

As we are interested in Liouvillians with local densities we focus on jump operators where the index $a$ runs either over the sites or the nearest-neighbour bonds of a one dimensional ring. In this setting $-iH - \sum_a \frac{\gamma_a}{2} L_a^\dagger L_a$ and $i\bar{H} - \sum_a \frac{\gamma_a}{2} \overline{L_a^\dagger L_a}$ describe interactions along the two legs of the ladder, while $\sum_a \gamma_a L_a \bar{L}_a^\dagger$ play the role of interactions between the two legs.

## 3.1 Single-site jump operators

In translationally invariant situations the most general bosonic single-site jump operator can be written in the form

$$\ell_j = \sum_{\alpha,\beta} \lambda_{\alpha\beta} E_j^{\alpha\beta} , \tag{31}$$

where $\lambda_{\alpha\beta} = 0$ unless $(\epsilon_\alpha + \epsilon_\beta) \bmod 2 = 0$. This generates "interaction terms" between the two legs of the form

$$\ell_j \overline{\ell_j^\dagger} = \sum_{\alpha\beta} \sum_{\gamma\delta} \lambda_{\alpha\beta} \, \lambda_{\gamma\delta}^* \, E_j^{\alpha\beta} \widetilde{E}_j^{\gamma\delta} . \tag{32}$$

The other jump operator terms in the Liouvillian generate "generalized magnetic field terms" acting on the two legs

$$\ell_j^\dagger \ell_j + \overline{\ell_j^\dagger \ell_j} = \sum_{\beta,\gamma} \Lambda_{\beta\gamma} E_j^{\beta\gamma} + \Lambda_{\gamma\beta} \widetilde{E}_j^{\beta\gamma} , \tag{33}$$

where $\Lambda_{\beta\gamma} = \sum_\alpha \lambda_{\alpha\beta}^* \lambda_{\alpha\gamma}$.

## 3.2 Single-bond jump operators

The most general bosonic jump operator acting on a bond takes the form

$$L_j = \sum_{\alpha,\beta} \lambda_{\alpha\beta} E_j^{\alpha\beta} + \lambda'_{\alpha\beta} E_{j+1}^{\alpha\beta} + \sum_{\alpha,\beta,\gamma,\delta} \mu_{\alpha\beta\gamma\delta}\, E_j^{\alpha\beta} E_{j+1}^{\gamma\delta}\,. \tag{34}$$

This gives rise to quartic, cubic and quadratic "interaction terms" in the Liouvillian. The resulting explicit expression is presented in Appendix A. The extension to fermionic jump operators is straightforward.

## 3.3 General form of the Liouvillian

In the following we will consider Liouvillians of the form

$$\mathcal{L} = -iH + i\bar{H} + \sum_{j=1}^{L}\sum_{a} \gamma_a \left[ L_j^{(a)}\overline{(L_j^{(a)})^\dagger} - \frac{1}{2}\left( (L_j^{(a)})^\dagger L_j^{(a)} + \overline{(L_j^{(a)})^\dagger L_j^{(a)}} \right) \right], \tag{35}$$

where $L_j^{(a)}$ are jump operators that act either on site $j$ or the bond $(j, j+1)$ and $\gamma_a > 0$. Our aim is to identify cases which are Yang-Baxter integrable. In practice this means that we need to check whether any of the large number of integrable Hamiltonians that can be interpreted as two-leg ladder models can be cast in the particular form (35). An added complication is that we should allow for general similarity transformations, i.e. consider

$$\mathcal{L}' = S\mathcal{L}S^{-1}\,. \tag{36}$$

The spatial locality of the Hamiltonian density of the various integrable models imposes strong restrictions on the possible form of $S$. Transformations of the form

$$S = \prod_{j=1}^{L} S_j\,, \tag{37}$$

where $S_j$ acts non-trivially only on site $j$ are always compatible with the aforementioned local structure.

# 4 Generalized Hubbard models

The first example of a Lindblad equation that is related to an interacting Yang-Baxter integrable model was presented in Ref. [27], where it was shown that the Lindblad equation for a tight-binding chain with dephasing noise can be mapped onto a fermionic Hubbard model with purely imaginary interactions. We now briefly review some results obtained in that work. We then show that the mathematical structure that underlies the integrability of the Hubbard model quite naturally leads to a connection with a Lindblad equation.

## 4.1 SU(2) Hubbard model

The Hubbard Hamiltonian is given by

$$H = -t\sum_{j=1}^{L}\sum_{\sigma=\uparrow,\downarrow} c_{j,\sigma}^\dagger c_{j+1,\sigma} + c_{j+1,\sigma}^\dagger c_{j,\sigma} + U\sum_{j=1}^{L}\left[ n_{j,\uparrow} - \frac{1}{2}\right]\left[ n_{j,\downarrow} - \frac{1}{2}\right], \tag{38}$$

where $n_{j,\sigma} = c_{j,\sigma}^\dagger c_{j,\sigma}$. The model is integrable for any complex value of $U/t$ [41]. In terms of the notations of section 2.1 we can choose a basis such that

$$c_{j,\uparrow}^\dagger = e_j^{21} \,, \quad c_{j,\downarrow}^\dagger = \widetilde{e}_j^{21} \,, \quad n_{j,\uparrow} = e_j^{22} \,, \quad n_{j,\downarrow} = \widetilde{e}_j^{22} \,, \tag{39}$$

and concomitantly

$$H(U) \;=\; -t \sum_j \Big[ e_j^{21} e_{j+1}^{12} + \widetilde{e}_j^{21} \widetilde{e}_{j+1}^{12} + \text{h.c.} \Big] + U \sum_j \Big[ e_j^{22} - \frac{1}{2} \Big] \Big[ \widetilde{e}_j^{22} - \frac{1}{2} \Big] \,. \tag{40}$$

### 4.1.1 Associated Lindblad equation

Let us consider a tight-binding model

$$H_0 = -t \sum_j e_j^{12} e_{j+1}^{21} + \text{h.c.} \,, \tag{41}$$

coupled to an environment by jump operators

$$L_j = e_j^{22} \,. \tag{42}$$

In the super-operator formalism the corresponding Liouvillian (22) is

$$\mathcal{L}(\gamma) = it \sum_j [e_j^{12} e_{j+1}^{21} - \widetilde{e}_j^{12} \widetilde{e}_{j+1}^{21} + \text{h.c.}] + \sum_j \gamma \Big[ e_j^{22} \widetilde{e}_j^{22} - \frac{1}{2} (e_j^{22} + \widetilde{e}_j^{22}) \Big] \,. \tag{43}$$

This is related to the Hubbard Hamiltonian by [27]

$$\mathcal{L}(\gamma) = -i\mathcal{U}^\dagger H(i\gamma)\mathcal{U} - \frac{\gamma L}{4} \,, \quad \mathcal{U} = \prod_{j=1}^{L/2} (\widetilde{e}_{2j}^{11} - \widetilde{e}_{2j}^{22}) \,. \tag{44}$$

## 4.2 Integrable structure of generalized Hubbard models and associated Lindblad equations

The Hubbard model was embedded into the general framework of the Quantum Inverse Scattering Method [40] in seminal work by Shastry [31,32]. This construction was subsequently generalized to other classes of integrable models [35–39]. The construction is based on an R-matrix $r_{12}(\lambda)$ acting on the tensor product of two graded linear vector spaces $V \otimes V$ and a conjugation matrix $C$ acting on $V$ that fulfil the Yang-Baxter relation

$$r_{12}(\lambda_1 - \lambda_2) r_{13}(\lambda_1 - \lambda_3) r_{23}(\lambda_2 - \lambda_3) = r_{23}(\lambda_2 - \lambda_3) r_{13}(\lambda_1 - \lambda_3) r_{12}(\lambda_1 - \lambda_2) \,, \tag{45}$$

as well as the "decorated" Yang-Baxter relation

$$r_{12}(\lambda_1 + \lambda_2) C_1 r_{13}(\lambda_1 - \lambda_3) r_{23}(\lambda_2 + \lambda_3) = r_{23}(\lambda_2 + \lambda_3) r_{13}(\lambda_1 - \lambda_3) C_1 r_{12}(\lambda_1 + \lambda_2) \,. \tag{46}$$

In the cases considered below the $r_{12}(\lambda)$ is given by

$$r_{12}(\lambda) = \Big[ \cos^2\Big(\frac{\lambda}{2}\Big) - \sin^2\Big(\frac{\lambda}{2}\Big) C_1 C_2 \Big] \Pi_{12} + \frac{\sin(\lambda)}{2} [\mathbb{I} \otimes \mathbb{I} - C_1 C_2] \,, \tag{47}$$

where $\Pi_{12}$ is a graded permutation operator (9) acting on $V \otimes V$ and

$$C = 2\hat{\pi} - \mathbb{1} \,, \tag{48}$$

where $\hat{\pi}$ is a projection operator onto a subspace of $V$. The R-matrix of an integrable generalized Hubbard model is then obtained by gluing together two copies [37, 39, 41]

$$
\begin{aligned}
R_{\langle 12\rangle\langle 34\rangle}(\lambda_1, \lambda_2) &= r_{13}(\lambda_1 - \lambda_2) r_{24}(\lambda_1 - \lambda_2) \\
&+ \alpha(\lambda_1, \lambda_2) r_{13}(\lambda_1 + \lambda_2) C_1 r_{24}(\lambda_1 + \lambda_2) C_2 .
\end{aligned}
\tag{49}
$$

Here the function $\alpha(\lambda, \mu)$ is given by

$$
\alpha(\lambda, \mu) = \frac{\cos(\lambda - \mu) \sinh\big(h(\lambda) - h(\mu)\big)}{\cos(\lambda + \mu) \cosh\big(h(\lambda) - h(\mu)\big)} ,
\tag{50}
$$

where $h(\mu)$ is a solution of the equation

$$
\sinh\big(2h(\lambda)\big) = U \sin(2\lambda) .
\tag{51}
$$

The local Hamiltonian density of the integrable "fundamental spin model" [40] corresponding to this R-matrix is

$$
\begin{aligned}
H_{\langle 12\rangle\langle 34\rangle} &= \left.\frac{d}{d\lambda}\right|_{\lambda = u_0} \Pi_{13}\Pi_{24} R_{\langle 12\rangle\langle 34\rangle}(\lambda, u_0) \\
&= \Pi_{13} r'_{13}(0) + \Pi_{24} r'_{24}(0) + \alpha'(u_0, u_0)\Pi_{13} r_{13}(2u_0) C_1 \Pi_{24} r_{24}(2u_0) C_2 .
\end{aligned}
\tag{52}
$$

Here we have generalized the construction of [37] by taking the logarithmic derivative of the transfer matrix at a shifted point $u_0$ following Ref. [43, 44]. Importantly the structure of the Hamiltonians (52) is such that they all can be related to Liouvillians of Lindblad equations. In the following we discuss a number of examples.

## 4.3 USW model

As a first application we consider eqn (52) for the case of the Hubbard model R-matrix [44]. The Hamiltonian of these models was first derived by Umeno, Shiroishi and Wadati in [43] and is of the form

$$
H_{\text{USW}}(U) = -\sum_j \left[ e_j^{21} e_{j+1}^{12} + \tilde{e}_j^{21} \tilde{e}_{j+1}^{12} + \text{h.c.} \right] + \frac{U}{\cosh\left(2h(u_0)\right)} \sum_j B_{j,j+1} \tilde{B}_{j,j+1} ,
\tag{53}
$$

where

$$
B_{j,j+1} = \left[ \cos^2(u_0)\left(e_j^{11} - e_j^{22}\right) - \sin^2(u_0)\left(e_{j+1}^{11} - e_{j+1}^{22}\right) + \sin(2u_0)\left(e_j^{21} e_{j+1}^{12} - e_{j+1}^{21} e_j^{12}\right) \right] ,
\tag{54}
$$

and $\tilde{B}_{j,j+1}$ is obtained from $B_{j,j+1}$ by replacing $e_n^{\alpha\beta} \to \tilde{e}_n^{\alpha\beta}$. Here $u_0$ is a free (complex) parameter and the function $h(u)$ is fixed by the requirement

$$
\sinh\left(2h(u_0)\right) = U \sin(2u_0) .
\tag{55}
$$

We note that the operators $e_j^{\alpha\beta}$ are related to spinful fermion creation and annihilation operators by (39). The Hamiltonian (53) is SO(4) symmetric [43] and in particular commutes with the total particle number

$$
\hat{N} = \sum_{j=1}^{L} e_j^{22} + \tilde{e}_j^{22} .
\tag{56}
$$

### 4.3.1 Associated Lindblad equation

The USW model is related to a Lindblad equation with a tight-binding Hamiltonian

$$H_0 = -\sum_j e_j^{12} e_{j+1}^{21} + \text{h.c.} \,, \tag{57}$$

and jump operators

$$L_j \;=\; B_{j,j+1} \,, \tag{58}$$

where the parameter $u_0$ is taken to be purely imaginary. In the super-operator formalism the corresponding Liouvillian (22) is

$$\mathcal{L}(\gamma) = i \sum_j \Big[ e_j^{12} e_{j+1}^{21} - \widetilde{e}_j^{12} \widetilde{e}_{j+1}^{21} + \text{h.c.} \Big] + \gamma \sum_j \Big[ B_{j,j+1} \tilde{B}_{j,j+1}^* - \cos^2(2u_0) \Big] \,. \tag{59}$$

This is related to the USW Hamiltonian by

$$\mathcal{L}(\gamma) = -i\mathcal{U}^\dagger H_{\text{USW}}(\mathfrak{u})\mathcal{U} - \gamma \cos^2(2u_0) L \,, \tag{60}$$

where the unitary transformation $\mathcal{U}$ is given by (44) and the parameter $\mathfrak{u}$ is purely imaginary and related to $\gamma$ by

$$\gamma = -i \frac{\mathfrak{u}}{\cosh\big(2h(u_0)\big)} \,. \tag{61}$$

### 4.3.2 Differential equations for correlation functions

As the jump operators are Hermitian the Lindblad equation implies the following time evolution for expectation values of (time independent) operators

$$\frac{d}{dt} \text{Tr}[\rho(t)\mathcal{O}] = -i\text{Tr}(\rho(t)[\mathcal{O}, H_0]) + \frac{\gamma}{2} \sum_j \text{Tr}\big(\rho(t)\,[[L_j, \mathcal{O}], L_j]\big) \,. \tag{62}$$

It is straightforward to verify that the jump operators (58) fulfil

$$\begin{aligned}
[L_n, c_j] \;=\;& 2\delta_{n,j-1} \sin(u_0)\big( \cos(u_0)c_{j-1} - \sin(u_0)c_j \big) \\
&+\; 2\delta_{n,j} \cos(u_0)\big( \cos(u_0)c_j - \sin(u_0)c_{j+1} \big) \,.
\end{aligned} \tag{63}$$

This shows that n-particle Green's functions fulfil simple, closed evolution equations. This is analogous to the case of the imaginary-U Hubbard model [27]. For example, the single-particle Green's function

$$G_{j,k}(t) = \text{Tr}\Big[\rho(t)c_j^\dagger c_k\Big] \tag{64}$$

has the following equation of motion

$$\begin{aligned}
\frac{d}{dt} G_{j,k} \;=\;& \sum_{\ell,m} K_{j,k}^{\ell,m} G_{\ell,m} \,, \\
K_{j,k}^{\ell,m} \;=\;& \delta_{j,\ell}\delta_{k-1,m}\Big[ i - \frac{\gamma \sin(4u_0)}{2} \Big] + \delta_{j,\ell}\delta_{k+1,m}\Big[ i + \frac{\gamma \sin(4u_0)}{2} \Big] \\
&-\; \delta_{j-1,\ell}\delta_{k,m}\Big[ i - \frac{\gamma \sin(4u_0)}{2} \Big] - \delta_{j+1,\ell}\delta_{k,m}\Big[ i + \frac{\gamma \sin(4u_0)}{2} \Big] \\
&-\; 4\gamma \delta_{j,\ell}\delta_{k,m} \cos^2(2u_0) - 4\gamma \delta_{j,k}\Big[ \sin^2(u_0) M_{j-1}^{\ell,m} - \cos^2(u_0) M_j^{\ell,m} \Big] \\
&-\; 4\gamma \Big[ \delta_{j-1,k} \sin(u_0)\cos(u_0) M_{j-1}^{\ell,m} - \delta_{j,k-1} \sin(u_0)\cos(u_0) M_j^{\ell,m} \Big] \,.
\end{aligned} \tag{65}$$

Here we have defined

$$M_j^{\ell,m} = \cos^2(u_0)\delta_{\ell,j}\delta_{m,j} - \sin(u_0)\cos(u_0)\big[\delta_{\ell,j}\delta_{m,j+1} - \delta_{\ell,j+1}\delta_{m,j}\big] - \sin^2(u_0)\delta_{\ell,j+1}\delta_{m,j+1} \,. \tag{66}$$

### 4.4 Maassarani models

In [33, 34] Maassarani introduced a class of integrable $2n$-state models that generalize the Hubbard model along the lines set out in section 4.2 above. We now discuss these models in more detail. A basis of the local Hilbert space is given by the tensor product

$$|a\rangle \otimes |\tilde{a}\rangle , \quad a, \tilde{a} = 1, \ldots, n , \tag{67}$$

where all states are bosonic, i.e. $\epsilon_a = 0 = \epsilon_{\tilde{a}}$. While these models a priori are generalized spin models they can be related to interacting fermion models by Jordan-Wigner transformations as is done for a simple case below. A basis of operators acting on these states is then given by $e_j^{ab} \tilde{e}_j^{\tilde{a}\tilde{b}}$. In terms of these (bosonic) operators Maassarani's Hamiltonian reads

$$H_{\text{Ma,n}}(U) = \sum_{j=1}^{L} P_{j,j+1}^{(n)} + \widetilde{P}_{j,j+1}^{(n)} + U\left(C_j \widetilde{C}_j - 1\right) , \tag{68}$$

where

$$
\begin{aligned}
P_{j,j+1}^{(n)} &= \sum_{a \in A} \sum_{b \in B} x_{ab} e_j^{ba} e_{j+1}^{ab} + x_{ab}^{-1} e_j^{ab} e_{j+1}^{ba} , \\
C_j &= \sum_{a \in A} e_j^{aa} - \sum_{b \in B} e_j^{bb} .
\end{aligned}
\tag{69}
$$

Here the two sets $A$ and $B$ form an arbitrary partition of $\{1, \ldots, n\}$ and $x_{ab}$ are arbitrary complex parameters. In the following we will simply set them equal to 1. The operators $\widetilde{P}_{j,j+1}^{(n)}$ and $\widetilde{C}_j$ are of the same forms as $P_{j,j+1}^{(n)}$ and $C_j$ respectively but with the replacement $e_j^{ab} \to \tilde{e}_j^{ab}$.

Maassarani's models are related to Lindblad equations with Hamiltonians

$$H_0^{(n)} = -\sum_j \left[ \sum_{a \in A} \sum_{b \in B} e_j^{ba} e_{j+1}^{ab} + e_j^{ab} e_{j+1}^{ba} \right] , \tag{70}$$

and jump operators

$$L_j = c - C_j , \tag{71}$$

where $c \in \mathbb{R}$ is a free parameter. In the superoperator formalism the corresponding Liouvillian is

$$\mathcal{L}_{\text{Ma,n}}(\gamma) = -i(H_0^{(n)} - \widetilde{H}_0^{(n)}) + \gamma \sum_j \left[ C_j \widetilde{C}_j - 1 \right] , \tag{72}$$

where $\widetilde{H}_0^{(n)}$ is of the same form as $H_0^{(n)}$ but with $e_j^{ab}$ replaced by $\tilde{e}_j^{ab}$. This is related to Maassarani's Hamiltonian by

$$\mathcal{L}_{\text{Ma,n}}(\gamma) = i\mathcal{U} H_{\text{Ma,n}}(-i\gamma) \mathcal{U}^\dagger , \quad \mathcal{U} = \prod_{j=1}^{L/2} \widetilde{C}_{2j} . \tag{73}$$

#### 4.4.1 3-state Maassarani model

The simplest Maassarani model is obtained by considering a local Hilbert space of three bosonic states. Choosing a decomposition $A = \{1\}$, $B = \{2, 3\}$ gives

$$H_0^{(3)} = -\sum_j e_j^{21} e_{j+1}^{12} + e_j^{31} e_{j+1}^{13} + \text{h.c.} . \tag{74}$$

In order to fermionize this model we embed it into an enlarged Hilbert space with four states per site, and then employ the results of section 2.1. This gives

$$e_j^{12} = \mathfrak{e}_j^{12}\,\widetilde{\mathfrak{e}}_j^{11}\,, \qquad e_j^{13} = \mathfrak{e}_j^{11}\widetilde{\mathfrak{e}}_j^{12}\,. \tag{75}$$

Finally we carry out a Jordan-Wigner transformation

$$\mathfrak{e}_j^{21} = \prod_{\ell=1}^{j-1}(1-2n_{\ell,\uparrow})c_{j,\uparrow}^\dagger\,, \qquad \widetilde{\mathfrak{e}}_j^{21} = \prod_{\ell=1}^{L}(1-2n_{\ell,\uparrow})\prod_{\ell=1}^{j-1}(1-2n_{\ell,\downarrow})c_{j,\downarrow}^\dagger\,. \tag{76}$$

After these transformations the Hamiltonian $H_0^{(3)}$ can be written in the form

$$H_0^{(3)} = -\mathcal{P}\sum_{j,\sigma}\Big[c_{j+1,\sigma}^\dagger c_{j,\sigma} + \text{h.c.}\Big]\mathcal{P}\,, \tag{77}$$

where

$$\mathcal{P} = \prod_{j=1}^{L}(1-n_{j,\uparrow}n_{j,\downarrow}) \tag{78}$$

is a projection operator that ensures that all sites are at most singly occupied. The Hamiltonian (77) can be viewed as the $U \to \infty$ limit of the Hubbard model and is sometimes referred to as the $t-0$ model. In terms of the fermionic operators the jump operator takes the form

$$L_j = 1 - 2(1-n_{j,\uparrow})(1-n_{j,\downarrow}) + c\,. \tag{79}$$

Choosing $c = 1$ we have

$$L_j|0\rangle = 0\,, \qquad L_j c_{j,\sigma}^\dagger|0\rangle = 2c_{j,\sigma}^\dagger|0\rangle\,, \tag{80}$$

which shows that the bath acts on the charge degrees of freedom. The Hamiltonian part $H_0^{(3)}$ has a free fermionic spectrum [45, 46], but the creation operators of the non-interacting fermion degrees of freedom are related to the $c_{j,\sigma}^\dagger$ in a non-local way [47, 48]. As a result the single-particle Green's function does not obey a simple evolution equation. The time evolution is again given by the general expression (62), where the relevant commutators are

$$\begin{aligned}
[[L_n, c_{j,\sigma}], L_n]\mathcal{P} &= -4c_{j,\sigma}\delta_{j,n}\,\mathcal{P}\,, \\
\mathcal{P}[c_{j,\sigma}, H_0^{(3)}]\mathcal{P} &= \mathcal{P}\big[-(c_{j+1,\sigma}+c_{j-1,\sigma}) - c_{j,\bar\sigma}^\dagger c_{j,\sigma}(c_{j+1,\bar\sigma}+c_{j-1,\bar\sigma})\big]\mathcal{P}\,.
\end{aligned} \tag{81}$$

### 4.4.2 4-state Maassarani model

In the 4-state case we can express the $e_j^{ab}$ in terms of two species of Pauli operators, *cf.* 2.1. Choosing $A = \{1, 2, 3\}$ and $B = \{4\}$ we then can interpret $H_0^{(4)}$ as the Hamiltonian of a two-leg spin ladder model

$$\begin{aligned}
H_0^{(4)} = \sum_{j=1}^{L}\Big[&\sigma_j^+\sigma_{j+1}^-\tau_j^+\tau_{j+1}^- + \sigma_j^-\sigma_{j+1}^+\tau_j^-\tau_{j+1}^+ + \frac{1}{4}(\sigma_j^+\sigma_{j+1}^- + \sigma_j^-\sigma_{j+1}^+)(1-\tau_j^z)(1-\tau_{j+1}^z) \\
&+\frac{1}{4}(\tau_j^+\tau_{j+1}^- + \tau_j^-\tau_{j+1}^+)(1-\sigma_j^z)(1-\sigma_{j+1}^z)\Big]\,.
\end{aligned} \tag{82}$$

The jump operators become (setting again $c = 1$ in (71))

$$L_j = \frac{1}{2}(1-\sigma_j^z)(1-\tau_j^z)\,. \tag{83}$$

### 4.4.3 Bethe Ansatz solution

The Maassarani models have been solved by Bethe Ansatz in Ref. [49]. Without loss of generality we restrict our discussion to the case where the sets $A$ and $B$ in (69) are given by

$$A = \{1, 2, \ldots, p\} , \qquad B = \{p+1, p+2, \ldots, n\} . \tag{84}$$

The exact eigenstates of $H_{\mathrm{Ma},n}(U)$ are then labelled by good quantum numbers as follows. The operators

$$Q^a = \sum_{j=1}^{L} e_j^{aa} , \qquad \tilde{Q}^a = \sum_{j=1}^{L} \tilde{e}_j^{aa} , \qquad a = 1, \ldots, n , \tag{85}$$

commute with $H_{\mathrm{Ma},n}(U)$ and with one another. Hence their eigenvalues $N_a, \tilde{N}_a$ can be used as good quantum numbers. Following Ref. [49] we introduce integers

$$N_A = \sum_{a=2}^{p} N_a , \quad N_B = \sum_{a=p+1}^{n} N_a , \quad \tilde{N}_A = \sum_{a=2}^{p} \tilde{N}_a , \quad \tilde{N}_B = \sum_{a=p+1}^{n-1} \tilde{N}_a , \tag{86}$$

and $N \geq N_A + N_B + \tilde{N}_A + \tilde{N}_B$. We then define sets

$$
\begin{aligned}
\mathcal{M}_A &= \{1, \ldots, N_A\} , \quad \mathcal{M}_B = \{N_A + 1, \ldots, N_A + N_B\} , \\
\tilde{\mathcal{M}}_A &= \{N_A + N_B + 1, \ldots, N_A + N_B + \tilde{N}_A\} , \\
\tilde{\mathcal{M}}_B &= \{N_A + N_B + \tilde{N}_A + 1, \ldots, N_A + N_B + \tilde{N}_A + \tilde{N}_B\} ,
\end{aligned}
\tag{87}
$$

and finally introduce two non-intersecting ordered sets of integers $1 \leq a_j \leq N \leq L$

$$\mathbb{A}_A = \{a_j | j \in \mathcal{M}_A\} , \quad \tilde{\mathbb{A}}_A = \{a_j | j \in \tilde{\mathcal{M}}_A\} , \quad \mathbb{A}_A \cap \tilde{\mathbb{A}}_A = \varnothing , \quad \mathbb{A}_A \cup \bar{\mathbb{A}}_A \equiv \mathbb{A} . \tag{88}$$

By ordered we mean that $a_j < a_{j+1}$ if $a_j, a_{j+1} \in \mathbb{A}_A$ and similarly for $\tilde{\mathbb{A}}_A$. The eigenstates of the Liouvillian $\mathcal{L}_{\mathrm{Ma},n}(\gamma)$ are then given in terms of rapidities $\{k_1, \ldots, k_N\}$, $\{\Lambda_j | j \in \mathcal{M}_B\}$, $\{b_m | m \in \tilde{\mathcal{M}}_B\}$ and integers $\{n_1, \ldots, n_{\tilde{N}_B - \tilde{N}_{n-1}}\}$, $\{\bar{n}_1, \ldots, \bar{n}_{N_B - N_n}\}$ subject to the following set of Bethe Ansatz equations [49]

$$e^{ik_j L} = e^{2\pi i \Phi} \prod_{l \in \mathcal{M}_B} \frac{\Lambda_l - \sin k_j + \gamma}{\Lambda_l - \sin k_j - \gamma} , \quad j \in [1, N] \backslash \mathbb{A} ,$$

$$\prod_{\substack{j=1 \\ j \notin \mathbb{A}}}^{N} \frac{\Lambda_m - \sin k_j + \gamma}{\Lambda_m - \sin k_j - \gamma} = e^{2\pi i \Psi} \prod_{\substack{l \in \mathcal{M}_B \\ l \neq m}} \frac{\Lambda_m - \Lambda_l + 2\gamma}{\Lambda_m - \Lambda_l - 2\gamma} , \quad m \in \mathcal{M}_B , \tag{89}$$

$$b_\ell^{\tilde{N}_B + \tilde{N}_n} = \prod_{j=1}^{\tilde{N}_B - \tilde{N}_{n-1}} e^{2\pi i \frac{n_j}{\tilde{N}_B}} , \quad 1 \leq n_1 < \cdots < n_{\tilde{N}_B - \tilde{N}_{n-1}} \leq \tilde{N}_B , \ \ell \in \tilde{M}_B ,$$

$$e^{ik_j(L - N_B)} = (-1)^{N_A - 1} e^{2\pi i \frac{m_\alpha}{N_A}} , \quad m_\alpha \in [1, N_A] , \quad j \in \mathbb{A}_A ,$$

$$e^{ik_j(L - \tilde{N}_B - \tilde{N}_n)} = (-1)^{\tilde{N}_A - 1} e^{2\pi i \frac{\tilde{m}_\alpha}{\tilde{N}_A}} , \quad \tilde{m}_\alpha \in [1, \tilde{N}_A] , \quad j \in \tilde{\mathbb{A}}_A , \tag{90}$$

where we require $\arg(b_\ell) < \arg(b_{\ell+1})$ and the phases $\Phi$ and $\Psi$ are given by

$$e^{2\pi i \Phi} = (-1)^{\tilde{N}_B + \tilde{N}_n - 1} \prod_{m \in \tilde{M}_B} b_m \prod_{j \in \tilde{M}_A} e^{-ik_{a_j}} ,$$

$$e^{2\pi i \Psi} = (-1)^{N - N_A - \tilde{N}_A} \prod_{j \in M_A} e^{-ik_{a_j}} \prod_{m \in \tilde{M}_A} e^{ik_{a_m}} \prod_{\ell \in \tilde{M}_B} b_\ell^{-1} \prod_{s=1}^{N_B - N_n} e^{2\pi i \frac{\bar{n}_s}{N_B}} ,$$

$$1 \leq \bar{n}_1 < \cdots < \bar{n}_{N_B - N_n} < N_B . \tag{91}$$

The corresponding eigenvalues of $\mathcal{L}_{\mathrm{Ma},n}(\gamma)$ are

$$E = 2i \sum_{j \in \mathcal{M}_B \cup \tilde{\mathcal{M}}_B} \cos k_j - 2\gamma(N_B + \tilde{N}_B + \tilde{N}_n) . \tag{92}$$

### 4.4.4 String solutions and vanishing of the Liouvillian gap in the thermodynamic limit

The first two sets (89) of the Bethe Ansatz equations are the same as for the Hubbard model with imaginary interactions strength and twisted boundary conditions. This ensures that the "$k$-$\Lambda$ string solutions" constructed in [27] are valid solutions for the $n$-state Maassarani models as well. A $k$-$\Lambda$ string of length $m$ corresponds to the following pattern of rapidities

$$
\begin{aligned}
k_{\alpha,j}^{(m)} &= \arcsin(i\Lambda_\alpha^{(m)} - (m - 2j + 2)\gamma') , \\
k_{\alpha,j+m}^{(m)} &= \pi - \arcsin(i\Lambda_\alpha^{(m)} + (m - 2j + 2)\gamma') , \\
\Lambda_{\alpha,j}^{(m)} &= i\Lambda_\alpha^{(m)} + \gamma(m + 1 - 2j) , \qquad 1 \le j \le m .
\end{aligned}
\tag{93}
$$

Here the string centres $\Lambda_\alpha^{(m)}$ are real and $\gamma' = -\gamma \, \mathrm{sgn}(\Lambda_\alpha^{(m)})$.

We now take $N_A = \tilde{N}_A = 0$ and consider a Bethe Ansatz state with a single $k$-$\Lambda$ string of length $m \ll L$. The corresponding eigenvalue of the Liouvillian is

$$\epsilon = 4\mathrm{Im}\sqrt{1 - (i|\Lambda_\alpha^{(m)}| - m\gamma)^2} - 4\gamma m . \tag{94}$$

In the framework of the string hypothesis the equation that fixes the allowed positions of the string centres $\Lambda_\alpha^{(m)}$ is obtained my "multiplying out the string" [50], which gives

$$\exp\left( iL \sum_{j=1}^{2m} k_j^{(m)} \right) = e^{2\pi im(2\Phi+\Psi)} . \tag{95}$$

Taking logarithms this can be cast in the form

$$\mathrm{sgn}(\Lambda^{(m)})\left[ \pi - \arcsin(i\Lambda^{(m)} + m\gamma) + \arcsin(i\Lambda^{(m)} - m\gamma) \right] = \frac{2\pi}{L}\left(J_\alpha^{(m)} + \varphi\right) , \tag{96}$$

where we have defined

$$\varphi = m(2\Phi + \Psi) \bmod 1 . \tag{97}$$

For even lattice lengths $L$ the $J_\alpha^{(m)}$ are integers with range

$$-\frac{L + 1 - 2m}{2} - \varphi < J^{(m)} < \frac{L + 1 - 2m}{2} - \varphi . \tag{98}$$

We now focus on the particular sequence of string states characterized by integers

$$J_\alpha^{(m)} = \frac{L}{2} - m - \alpha , \quad \alpha = 1, 2, \cdots \ll L . \tag{99}$$

In the limit of large system sizes $L \gg 1$ the corresponding string centres follow from (96)

$$\Lambda_\alpha^{(n)} = \frac{m\gamma L}{\pi(m + \alpha - \varphi)} + \mathcal{O}(1) . \tag{100}$$

Substituting this into our expression (94) for the eigenvalue of the Liouvillian gives

$$\epsilon_\alpha^{(m)} = -\frac{2\pi^2}{m\gamma L^2}(m + \alpha - \varphi)^2 + \mathcal{O}(L^{-4}) . \tag{101}$$

This shows that in the large-$L$ limit we have a band of Liouvillian eigenstates with eigenvalues that scale as $L^{-2}$. This establishes that the Liouvillian gap vanishes in the thermodynamic limit. Moreover, the scaling with system size suggests that the corresponding eigenmodes are diffusive. Our calculation does not rule out the existence of eigenstates with gaps that approach zero faster than $L^{-2}$.

### 4.5 $GL(N, M)$ **Maassarani models**

As we already mentioned above in section 4.2 the Shastry-Maassarani construction can be generalized to graded magnets based on $GL(N, M)$. Following Ref. [37] we consider the class of Hamiltonians

$$H_{\text{gMa}}(U) = \sum_j \Pi^{(n)}_{j,j+1} + \widetilde{\Pi}^{(n)}_{j,j+1} + U\left[C_j \widetilde{C}_j - 1\right], \tag{102}$$

where

$$\begin{aligned}
\Pi^{(n)}_{j,j+1} &= \sum_{k \neq N, K}\left[E_j^{kN} E_{j+1}^{Nk} - E_j^{kK} E_{j+1}^{Kk} + (-1)^{\epsilon_k}(E_j^{Nk} E_{j+1}^{kN} + E_j^{Kk} E_{j+1}^{kK})\right], \\
C_j &= 1 - 2E_j^{KK} - 2E_j^{NN}, \qquad K = N + M.
\end{aligned} \tag{103}$$

We can relate this to a Lindblad equation with Hamiltonian

$$H_0 = -\sum_j \Pi^{(n)}_{j,j+1}, \tag{104}$$

and jump operators

$$L_j = 1 - C_j. \tag{105}$$

#### 4.5.1 3-state $GL(1, 2)$ **model**

The simplest example is the 3-state model based in $GL(1, 2)$. Like in the case of the 3-state Maassarani model considered above we may represent the Hamiltonian in terms of canonical spinful fermion creation and annihilation operators by identifying the three states per site as

$$|1\rangle_j = |0\rangle_j, \quad |2\rangle_j = c^\dagger_{j,\uparrow}|0\rangle_j, \quad |3\rangle_j = c^\dagger_{j,\downarrow}|0\rangle_j. \tag{106}$$

Then $H_0$ can be represented as

$$H_0 = -\mathcal{P}\sum_{j=1}^{L}\left(c^\dagger_{j,\uparrow} c_{j+1,\uparrow} - S_j^+ S_{j+1}^- + \text{h.c.}\right)\mathcal{P}, \tag{107}$$

where $\mathcal{P}$ is the projection operator on singly occupied sites (78) and $S_j^+ = c^\dagger_{i,\uparrow} c_{j,\downarrow}$. This describes correlated hopping of the up fermions, whereas the down fermions can only move through spin-flip processes. The jump operator is

$$L_j = 2n_{j,\uparrow} - 1. \tag{108}$$

## 5 Other integrable two-leg ladder models

The generalized Hubbard models considered above are all related to Lindblad equations with a single jump operator on each bond by virtue of their integrability structure. There are many other integrable models that can be represented as two-leg ladders and a question we have investigated at some length is whether some of them can be associated with Lindblad equations as well.

## 5.1 $GL(N^2)$ magnets

We now consider generalized spin models on a local Hilbert space with $N^2$ bosonic states. A well-known class of integrable models is obtained by taking [51, 52]

$$H_{GL(N^2)} = \sum_{j=1}^{L} \sum_{\alpha,\beta=1}^{N^2} E_j^{\alpha\beta} E_{j+1}^{\beta\alpha} , \tag{109}$$

where $P_{j,j+1} = \sum_{\alpha,\beta=1}^{N^2} E_j^{\alpha\beta} E_{j+1}^{\beta\alpha}$ is a permutation operator acting on nearest-neighbour lattice sites

$$P_{j,j+1}|\gamma\rangle_j|\delta\rangle_{j+1} = |\delta\rangle_j|\gamma\rangle_{j+1} . \tag{110}$$

The Hamiltonian $H$ is $GL(N^2)$ symmetric and hence

$$[H, Q^{\alpha,\beta}] = 0 , \quad Q^{\alpha,\beta} = \sum_{j=1}^{L} E_j^{\alpha\beta} . \tag{111}$$

### 5.1.1 Representation as a 2-leg ladder

The permutation models can be viewed as 2-leg ladders by employing the decomposition of section 2.1 for $M = N$. This provides a representation of the permutation operator as a tensor product

$$P_{j,j+1} = \left[ \sum_{\alpha,\beta=1}^{N} \widetilde{e}_j^{\alpha\beta} \widetilde{e}_{j+1}^{\beta\alpha} \right] \left[ \sum_{\gamma,\delta=1}^{N} e_j^{\gamma\delta} e_{j+1}^{\delta\gamma} \right] . \tag{112}$$

It is clear from the representation (112) that

$$[H, J^{\alpha\beta}] = 0 = [H, \widetilde{J}^{\alpha\beta}] , \tag{113}$$

where

$$\widetilde{J}^{\alpha\beta} = \sum_{j=1}^{L} \widetilde{e}_j^{\alpha\beta} , \quad J^{\alpha\beta} = \sum_{j=1}^{L} e_j^{\alpha\beta} , \qquad \alpha, \beta = 1, \ldots N . \tag{114}$$

These operators are related to the $GL(N^2)$ symmetry generators by

$$J^{\alpha\beta} = \sum_{\gamma=1}^{N} Q^{N(\gamma-1)+\alpha, N(\gamma-1)+\beta} , \quad \widetilde{J}^{\alpha\beta} = \sum_{\gamma=1}^{N} Q^{N(\alpha-1)+\gamma, N(\beta-1)+\gamma} . \tag{115}$$

### 5.1.2 Associated Lindblad equation

Consider now a Lindblad equation with Hamiltonian $H_0$ and two sets of jump operators $\{L_j\}$ and $\{\ell_j^{\alpha\beta}\}$

$$H_0 = \sum_{\alpha,\beta=1}^{N} \lambda_{\alpha\beta} J^{\alpha\beta} , \quad L_j = \left[ \sum_{\bar{\alpha},\bar{\beta}=1}^{N} e_j^{\bar{\alpha}\bar{\beta}} e_{j+1}^{\bar{\beta}\bar{\alpha}} \right] , \quad \ell_j^{\alpha\beta} = e_j^{\alpha\beta} . \tag{116}$$

Noting that $L_j^\dagger L_j = \mathbb{1}$ we conclude that the corresponding Liouvillian is

$$\mathcal{L} = \sum_{\alpha,\beta=1}^{N^2} f_{\alpha\beta} Q^{\alpha,\beta} + \gamma \sum_{j=1}^{L} \left( P_{j,j+1} - 1 \right) , \tag{117}$$

where

$$\sum_{\alpha,\beta=1}^{N^2} f_{\alpha\beta} Q^{\alpha,\beta} = \sum_{\bar\alpha,\bar\beta=1}^{N} -i\lambda_{\bar\alpha\bar\beta} [J^{\bar\alpha\bar\beta} - \widetilde{J}^{\bar\alpha\bar\beta}] + \gamma_{\bar\alpha\bar\beta} \left[ Q^{N(\bar\alpha-1)+\bar\alpha, N(\bar\beta-1)+\bar\beta} - \frac{J^{\bar\beta\bar\beta} + \widetilde{J}^{\bar\beta\bar\beta}}{2} \right]. \quad (118)$$

By construction the first term in (117) commutes with the second, which is $\gamma H_{GL(N^2)}$. As $H_{GL(N^2)}$ is invariant under all global $GL(N^2)$ rotations $U$ we conclude that (117) is integrable for choices of $\gamma_{\alpha\beta}$ and $\lambda_{\alpha\beta}$ such that

$$U \sum_{\alpha,\beta=1}^{N^2} f_{\alpha\beta} Q^{\alpha,\beta} U^\dagger = \sum_{\alpha=1}^{N^2} g_\alpha Q^{\alpha,\alpha}, \quad g_\alpha \in \mathbb{C}. \quad (119)$$

### 5.1.3 Twisting the boundary conditions

As we have mentioned above, in general we need to consider similarity transformations when trying to ascertain whether a Lindblad equation is related to an integrable Hamiltonian. A simple example is provided by considering a Lindblad equation with vanishing Hamiltonian and jump operators

$$L_j = \left[ \sum_{\bar\alpha,\bar\beta=1}^{N} e^{i(\varphi_{\bar\beta}-\varphi_{\bar\alpha})} e_j^{\bar\alpha\bar\beta} e_{j+1}^{\bar\beta\bar\alpha} \right], \quad \varphi_{\bar\alpha} \in \mathbb{R}. \quad (120)$$

The corresponding Liouvillian is

$$\begin{aligned}
\mathcal{L} &= \gamma \sum_{j=1}^{L} \left[ \sum_{\widetilde\alpha,\widetilde\beta,\bar\alpha,\bar\beta=1}^{N} e^{i(\varphi_{\bar\beta}-\varphi_{\bar\alpha}-\varphi_{\widetilde\beta}+\varphi_{\widetilde\alpha})} e_j^{\bar\alpha\bar\beta} e_{j+1}^{\bar\beta\bar\alpha} \widetilde{e}_j^{\widetilde\alpha\widetilde\beta} \widetilde{e}_{j+1}^{\widetilde\beta\widetilde\alpha} - 1 \right] \\
&= \gamma \sum_{j=1}^{L} \sum_{\alpha,\beta=1}^{N^2} \left[ E_j^{\alpha\beta} E_{j+1}^{\beta\alpha} e^{i(\phi_\beta-\phi_\alpha)} - 1 \right],
\end{aligned} \quad (121)$$

where we have used the decomposition 2.1 and fixed the phases $\phi_\alpha$ by

$$\phi_\beta - \phi_\alpha = \varphi_{\bar\beta} - \varphi_{\bar\alpha} + \varphi_{\widetilde\alpha} - \varphi_{\widetilde\beta}, \quad (122)$$

where $\alpha, \beta, \bar\alpha, \bar\beta, \widetilde\alpha, \widetilde\beta$ are related by (12). To relate this to the $GL(N^2)$ Hamiltonian we consider the canonical transformation

$$U E_j^{\alpha\beta} U^\dagger = E_j^{\alpha\beta} e^{-i(\phi_\alpha-\phi_\beta)j}, \quad (123)$$

under which the Liouvillian transforms as

$$U\mathcal{L}U^\dagger = \sum_{j=1}^{L} \sum_{\alpha,\beta=1}^{N^2} \left[ E_j^{\alpha\beta} E_{j+1}^{\beta\alpha} - 1 \right], \quad (124)$$

where we have imposed twisted boundary conditions

$$E_{L+1}^{\beta\alpha} = E_1^{\beta\alpha} e^{-i(\phi_\alpha-\phi_\beta)L}. \quad (125)$$

We conclude that the Liouvillian is related to the integrable $GL(N^2)$ Hamiltonian with twisted boundary conditions

$$U\mathcal{L}U^\dagger = \gamma H_{GL(N^2)} \Big|_{\text{twisted bc}}. \quad (126)$$

The integrability of twisted boundary conditions in the $GL(N^2)$ models is well known [53–55].

### 5.1.4 Example: GL(4) spin ladder

As a specific example let us consider the $GL(4)$ case

$$H = J \sum_{j=1}^{L} P_{j,j+1} + \frac{h}{2} \left[ Q^{1,1} + Q^{2,3} + Q^{3,2} + Q^{4,4} \right] , \tag{127}$$

where we have added a particular generalized magnetic field term. Using 2.1 we can express this in terms of two species of Pauli operators, *cf.* [42]

$$H = \frac{1}{4} \sum_{j=1}^{L} J \left( \sigma_j . \sigma_{j+1} + 1 \right) \left( \tau_j . \tau_{j+1} + 1 \right) + h \left( \sigma_j . \tau_j + 1 \right) . \tag{128}$$

The related Lindblad equation has no Hamiltonian and two sets of jump operators

$$L_j = \frac{1}{2} \sum_{a=x,y,z} \sigma_j^a \sigma_{j+1}^a + 1 , \quad \{ \ell_j^{(a)} = \sigma_j^a | a = x, y, z \} . \tag{129}$$

The corresponding Liouvillian is

$$\mathcal{L} = \gamma \sum_{j=1}^{L} \left( P_{j,j+1} - 1 \right) + \gamma' \sum_{j=1}^{L} ( \sigma_j^x \widetilde{\sigma}_j^x - \sigma_j^y \widetilde{\sigma}_j^y + \sigma_j^z \widetilde{\sigma}_j^z - 3 ) . \tag{130}$$

After a local basis rotation around the y-axis

$$\tau_j^x = -\widetilde{\sigma}_j^x , \quad \tau_j^y = \widetilde{\sigma}_j^y , \quad \tau_j^z = -\widetilde{\sigma}_j^z , \tag{131}$$

this maps onto (128) (up to a constant contribution) if we identify $\gamma = J/4$ and $h = -\gamma'$.

## 5.2 $GL(n_B^2 + n_F^2 | 2n_B n_F)$ magnets

We now turn to particular graded magnets, where we have $n_B^2 + n_F^2$ bosonic and $2n_B n_F$ fermionic states at a given site of the lattice, where $n_{B,F} \in \mathbb{N}_0$. A much studied family of integrable models is given by [52, 56–61]

$$H = \sum_j \Pi_{j,j+1} + \sum_j \sum_\alpha \lambda_\alpha E_j^{\alpha\alpha} , \tag{132}$$

where $\Pi_{j,j+1}$ is a graded permutation operator (9) and $\lambda_\alpha$ are generalized chemical potentials. The case $n_B = n_F = 1$ gives the EKS model (a.k.a. supersymmetric extended Hubbard model). We now employ the decomposition 2.1 and choose a tensor product basis for the local Hilbert space as

$$|\alpha\rangle = |\widetilde{\alpha}\rangle \otimes |\bar{\alpha}\rangle , \quad \epsilon_\alpha = \epsilon_{\widetilde{\alpha}} + \epsilon_{\bar{\alpha}} , \quad \alpha, \bar{\alpha} = 1, \dots, n_B + n_F , \tag{133}$$

where $\alpha = (n_B + n_F)(\widetilde{\alpha} - 1) + \bar{\alpha}$. The $E_j^{\alpha\beta}$'s can then be expressed as

$$E_j^{\alpha\beta} = (-1)^{\epsilon_{\bar{\beta}}(\epsilon_{\bar{\alpha}} + \epsilon_{\bar{\beta}})} \widetilde{e}_j^{\widetilde{\alpha}\widetilde{\beta}} e_j^{\bar{\alpha}\bar{\beta}} , \tag{134}$$

which in turn leads to the following decomposition of the graded permutation operator

$$\Pi_{j,j+1} = \left[ \sum_{\widetilde{\alpha},\widetilde{\beta}=1}^{n_B+n_F} (-1)^{\epsilon_{\widetilde{\beta}}} \widetilde{e}_j^{\widetilde{\alpha}\widetilde{\beta}} e_{j+1}^{\widetilde{\beta}\widetilde{\alpha}} \right] \left[ \sum_{\bar{\alpha},\bar{\beta}=1}^{n_B+n_F} (-1)^{\epsilon_{\bar{\beta}}} e_j^{\bar{\alpha}\bar{\beta}} e_{j+1}^{\bar{\beta}\bar{\alpha}} \right] . \tag{135}$$

### 5.2.1 Associated Lindblad equation

Consider now a Lindblad equation with no Hamiltonian and Hermitian jump operators

$$L_j = \sum_{\bar{\alpha},\bar{\beta}=1}^{N} (-1)^{\epsilon_{\bar{\beta}}} e_j^{\bar{\alpha}\bar{\beta}} \, e_{j+1}^{\bar{\beta}\bar{\alpha}} \, . \tag{136}$$

Noting that

$$L_j^\dagger L_j = \mathbb{1} \, , \tag{137}$$

we conclude that the corresponding Liouvillian is

$$\mathcal{L} = \gamma \sum_{j=1}^{L} (\Pi_{j,j+1} - 1) \, . \tag{138}$$

We can slightly generalize this by following the construction for the $GL(N^2)$ case, e.g. we can add a Hamiltonian

$$H = \sum_{\bar{\alpha}=1}^{n_B+n_F} \lambda_{\bar{\alpha}} \sum_{j=1}^{L} e_j^{\bar{\alpha}\bar{\alpha}} \, . \tag{139}$$

## 5.3 Integrable spin ladder model of Refs [62, 63]

The Hamiltonian of this model can be cast in the form of a two-leg spin ladder [42]

$$\begin{aligned}
H(J) &= \frac{1}{4} \sum_{j=1}^{L} \Big[ \big( \sigma_j.\sigma_{j+1} + 1 \big)\big( \tau_j.\tau_{j+1} + 1 \big) + J\big( \sigma_j.\tau_j + 1 \big) \\
&\quad + \big( \sigma_j.\tau_j + 1 \big)\big( \sigma_{j+1}.\tau_{j+1} + 1 \big) - \big( \sigma_j.\tau_{j+1} + 1 \big)\big( \tau_j.\sigma_{j+1} + 1 \big) \Big] \, .
\end{aligned} \tag{140}$$

### 5.3.1 Associated Lindblad equation

The Hamiltonian (140) is related to a Lindblad equation with no Hamiltonian part and a set of Hermitian jump operators

$$L_j = \sigma_j \cdot \sigma_{j+1} + 1 \, , \quad A_j^{(a)} = \sum_{b,c} \epsilon_{abc} \sigma_j^b \sigma_{j+1}^c \, , \quad B_j^{(a)} = \sigma_j^a + \sigma_{j+1}^a \, , \quad a = x,y,z \, . \tag{141}$$

After a local basis rotation

$$\tau_j^a \to \tau_j^y \tau_j^a \tau_j^y \, , \quad a = x,y,z \, , \tag{142}$$

and setting the $\gamma$ parameters to be equal for all jump operator terms we arrive at a Liouvillian

$$\mathcal{L} = 4\gamma H(-4) - 12L\gamma \, . \tag{143}$$

# 6 A comment on scaling limits

A standard way of generating integrable QFTs is by taking appropriate scaling limits of integrable lattice models. A paradigmatic example is the scaling limit of the Hubbard model, which gives rise to the integrable Yang-Gaudin model. An interesting question is whether we can carry out an analogous construction for our integrable Liouvillians and arrive at non-unitary integrable QFTs. The answer seems to be negative. Let us consider a lattice model with Hamiltonian

$$H_0 = -t \sum_j c_j^\dagger c_{j+1} + c_{j+1}^\dagger c_j - \mu \sum_j c_j^\dagger c_j \, , \tag{144}$$

where $c_j$ and $c_j^\dagger$ are annihilation and creation operators of spinless fermions, and jump operators

$$L_j = n_j = c_j^\dagger c_j \ . \tag{145}$$

These give rise to a Liouvillian

$$\mathcal{L} \ = \ -iH_0 + i\widetilde{H}_0 + \gamma \sum_j n_j \tilde{n}_j - \frac{1}{2}(n_j + \tilde{n}_j) \ , \tag{146}$$

where $\tilde{H}_0$ is of the same form as $H_0$ but written in terms of fermion annihilation and creation operators $\tilde{c}_j$ and $\tilde{c}_j^\dagger$. The sign difference between $\tilde{H}_0$ and $H_0$ can be removed by a canonical transformation

$$\tilde{c}_j \rightarrow \tilde{c}_j (-1)^j \ . \tag{147}$$

In analogy of what we do in order to obtain the Yang-Gaudin model from the Hubbard model we now consider the scaling limit

$$t \rightarrow \infty \ , \quad a_0 \rightarrow 0 \ , \quad t a_0^2 \text{ fixed.} \tag{148}$$

In this limit lattice fermion operators are replaced by continuum fields

$$c_j \simeq \sqrt{a_0} \Psi_\uparrow(x) \ , \quad \tilde{c}_j \simeq \sqrt{a_0} \Psi_\downarrow(x) \ , \quad x = j a_0 \ . \tag{149}$$

The Liouvillian becomes

$$
\begin{aligned}
\mathcal{L} \ = \ & \left[ i(2t + \mu) - \frac{\gamma}{2} \right] \int dx \sum_\sigma \Psi_\sigma^\dagger(x) \Psi_\sigma(x) \\
& + \ it a_0^2 \int dx \sum_\sigma \Psi_\sigma^\dagger(x) \partial_x^2 \Psi_\sigma(x) + \gamma a_0 \int dx \Psi_\uparrow^\dagger(x) \Psi_\uparrow(x) \Psi_\downarrow^\dagger(x) \Psi_\downarrow(x) \ .
\end{aligned}
\tag{150}
$$

A problem now occurs in the first term. If $\gamma$ were purely imaginary, as is the case for the Hubbard model, we could tune the chemical potential in such a way to ensure that the prefactor remains finite in the scaling limit. This would leave us with a Yang-Gaudin model at a finite density, but with imaginary interaction strength. However, given that $\gamma$ is real and positive we cannot take $\gamma \rightarrow \infty$, but must keep it finite in order to describe states with finite real parts of their "energies". This means that the only scaling limit is trivial as the interaction term disappears. This would appear to be a more general feature, independent of integrability.

# 7 Some unsuccessful maps

Most of the integrable ladder models we have considered cannot be associated in a straightforward way with Lindblad equations. In the following we present some representative examples.

## 7.1 Perk-Schultz models

As an example we consider the $N = 4$ Perk-Schultz model [64, 65]

$$H_{\mathrm{PS}} = J \sum_j \left[ \cosh(\eta) \sum_\alpha E_j^{\alpha\alpha} E_{j+1}^{\alpha\alpha} + \sum_{\alpha \neq \beta} E_j^{\beta\alpha} E_{j+1}^{\alpha\beta} + \mathrm{sgn}(\alpha - \beta) \sinh\eta \ E_j^{\alpha\alpha} E_{j+1}^{\beta\beta} \right] . \tag{151}$$

This can be viewed as a q-deformation of the GL(4) Hamiltonian considered above. Using the decomposition 2.1 we can rewrite $H_{\text{PS}}$ as

$$
\begin{aligned}
H_{\text{PS}} &= J\sum_j P_{j,j+1} + \frac{\cosh(\eta)-1}{4}\left(1+\sigma_j^z\sigma_{j+1}^z\right)\left(1+\tau_j^z\tau_{j+1}^z\right) \\
&\quad + \frac{J\sinh(\eta)}{4}\sum_j\left(\sigma_{j+1}^z-\sigma_j^z\right)\left(1+\tau_j^z\tau_{j+1}^z\right).
\end{aligned}
\tag{152}
$$

As the spectra of $\sigma_{j+1}^z-\sigma_j^z$ and $1+\tau_j^z\tau_{j+1}^z$ are different the term in the second line cannot be related to a jump operator structure in this representation.

## 7.2 Higher conservation laws

A well-known way of obtaining integrable spin-ladder models is by considering higher conservation laws [40, 66]. In case of the spin-1/2 Heisenberg XXX chain higher conservation laws $H^{(k+1)}$ can be obtained from the transfer matrix by taking logarithmic derivatives at the "shift point". By construction we have $[H^{(k)}, H^{(l)}] = 0$. The Hamiltonian we want to consider here is $H(b) = H^{(2)} + bH^{(4)} + \text{const}$ [66–68], which takes the form

$$
\begin{aligned}
H(b) &= 4\sum_{j=1}^{L}\Big[(1-b)\mathbf{S}_j\cdot\mathbf{S}_{j+1} + \frac{b}{2}\mathbf{S}_j\cdot\mathbf{S}_{j+2} + 2b\left(\mathbf{S}_{j-1}\cdot\mathbf{S}_{j+1}\right)\left(\mathbf{S}_j\cdot\mathbf{S}_{j+2}\right) \\
&\quad - 2b\left(\mathbf{S}_{j-1}\cdot\mathbf{S}_{j+2}\right)\left(\mathbf{S}_j\cdot\mathbf{S}_{j+1}\right)\Big].
\end{aligned}
\tag{153}
$$

This can be viewed as a zig-zag ladder model by associating all even (odd) sites with the first (second) leg, which gives

$$
\begin{aligned}
H(b) &= \sum_{j=1}^{L/2}(1-b)\boldsymbol{\sigma}_j\cdot\left[\boldsymbol{\tau}_j+\boldsymbol{\tau}_{j+1}\right] + \frac{b}{2}\Big\{\boldsymbol{\sigma}_j\cdot\boldsymbol{\sigma}_{j+1}\left[\boldsymbol{\tau}_j\cdot\boldsymbol{\tau}_{j+1}+\boldsymbol{\tau}_{j+1}\cdot\boldsymbol{\tau}_{j+2}\right] \\
&\quad + \boldsymbol{\sigma}_j\cdot\boldsymbol{\sigma}_{j+1}+\boldsymbol{\tau}_j\cdot\boldsymbol{\tau}_{j+1}-\boldsymbol{\tau}_j\cdot\boldsymbol{\sigma}_{j+1}\,\boldsymbol{\sigma}_j\cdot\boldsymbol{\tau}_{j+1}-\boldsymbol{\sigma}_j\cdot\boldsymbol{\tau}_{j+2}\,\boldsymbol{\sigma}_{j+1}\cdot\boldsymbol{\tau}_{j+1}\Big\}.
\end{aligned}
\tag{154}
$$

This is asymmetric under leg exchange in a way that precludes a direct relation with a Lindblad equation.

## 7.3 Alcaraz-Bariev model

The Alcaraz-Bariev two-parameter families of integrable models [69] come in two classes denoted by $A^\pm$ and $B^\pm$ respectively. The $B^\pm$ family contains the Hubbard model as a special limit and this is the only case in which we succeeded in obtaining an interpretation in terms of a Lindblad equation. We now discuss why such a relation does not seem to exist in general for the $A^\pm$ family of models. The Hamiltonian of the $A^\pm$ family can be cast in the form

$$
H_A^{(\epsilon)} = \sum_j T_{j,j+1} + T_{j,j+1}^{(1)} + T_{j,j+1}^{(2)} + gT_{j,j+1}^{(3)} + \cos\theta\left[S_{j,j+1} - \epsilon T_{j,j+1}^{(p)} + V_{j,j+1} - \epsilon U_{j,j+1}\right], \tag{155}
$$

where $g = (1 + \epsilon)(1 - \sin\theta)$ and

$$
\begin{aligned}
T_{j,j+1} &= -e_j^{21}e_{j+1}^{12} + \widetilde{e}_j^{21}\widetilde{e}_{j+1}^{12} + \text{h.c.}, \\
T_{j,j+1}^{(1)} &= -(e_j^{21}e_{j+1}^{12} - e_j^{12}e_{j+1}^{21})\widetilde{e}_j^{22}(\epsilon\sin\theta - 1) + (\widetilde{e}_j^{21}\widetilde{e}_{j+1}^{12} - \widetilde{e}_j^{12}\widetilde{e}_{j+1}^{21})e_j^{22}(\sin\theta - 1), \\
T_{j,j+1}^{(2)} &= -(e_j^{21}e_{j+1}^{12} - e_j^{12}e_{j+1}^{21})\widetilde{e}_{j+1}^{22}(\sin\theta - 1) + (\widetilde{e}_j^{21}\widetilde{e}_{j+1}^{12} - \widetilde{e}_j^{12}\widetilde{e}_{j+1}^{21})e_{j+1}^{22}(\epsilon\sin\theta - 1), \\
T_{j,j+1}^{(3)} &= -e_j^{21}e_{j+1}^{12}\widetilde{e}_j^{22}\widetilde{e}_{j+1}^{22} + \widetilde{e}_j^{21}\widetilde{e}_{j+1}^{12}e_j^{22}e_{j+1}^{22} + \text{h.c.}, \\
S_{j,j+1} &= e_j^{21}e_{j+1}^{12}\widetilde{e}_j^{12}\widetilde{e}_{j+1}^{21} + \text{h.c.}, \\
T_{j,j+1}^{(p)} &= -e_j^{21}e_{j+1}^{12}\widetilde{e}_j^{21}\widetilde{e}_{j+1}^{12} + \text{h.c.}, \\
V_{j,j+1} &= e^{-2\eta}e_j^{22}\widetilde{e}_{j+1}^{22} + e^{2\eta}\widetilde{e}_j^{22}e_{j+1}^{22}, \\
U_{j,j+1} &= e_j^{22}\widetilde{e}_j^{22} + e_{j+1}^{22}\widetilde{e}_{j+1}^{22}.
\end{aligned}
\tag{156}
$$

Here we have carried out a unitary transformation

$$
Ue_j^{ab}U^\dagger = e_j^{ab}(-1)^{j(a-b)}
\tag{157}
$$

on the Hamiltonian given in [69] in anticipation of relating it to a Liouvillian on a Lindblad equation. We start by noting that we require $g = 0$ for such an interpretation to be possible. The reason is that the only way to generate $T_{j,j+1}^{(3)}$ is as a "cross-term" in $\ell_j\overline{\ell}_j^\dagger$ with

$$
\ell_j = ae_j^{21}e_{j+1}^{12} + be_j^{12}e_{j+1}^{21} + ce_j^{22}e_{j+1}^{22}.
\tag{158}
$$

However, such jump operators would also generate an unwanted contribution

$$
|c|^2e_j^{22}e_{j+1}^{22}\widetilde{e}_j^{22}\widetilde{e}_{j+1}^{22}.
\tag{159}
$$

As this cannot be cancelled by introducing additional jump operators and does not feature in $H_A^{(\epsilon)}$ we conclude that we must have $g = 0$. Next we turn to the cubic terms $T_{j,j+1}^{(1)}$. These must arise from jump operators of the form

$$
L_j = ae_j^{21}e_{j+1}^{12} + be_j^{12}e_{j+1}^{21} + ce_j^{22}.
\tag{160}
$$

These jump operators give rise to inter-species interactions

$$
\begin{aligned}
L_j\overline{L}_j^\dagger &= |a|^2e_j^{21}e_{j+1}^{12}\widetilde{e}_j^{21}\widetilde{e}_{j+1}^{12} + ab^*e_j^{21}e_{j+1}^{12}\widetilde{e}_j^{12}\widetilde{e}_{j+1}^{21} + a^*be_j^{12}e_{j+1}^{21}\widetilde{e}_j^{21}\widetilde{e}_{j+1}^{12} \\
&\quad + |b|^2e_j^{12}e_{j+1}^{21}\widetilde{e}_j^{12}\widetilde{e}_{j+1}^{21} + |c|^2e_j^{22}\widetilde{e}_j^{22} + c^*(ae_j^{21}e_{j+1}^{12} + be_j^{12}e_{j+1}^{21})\widetilde{e}_j^{22} \\
&\quad + ce_j^{22}(a^*\widetilde{e}_j^{21}\widetilde{e}_{j+1}^{12} + b^*\widetilde{e}_j^{12}\widetilde{e}_{j+1}^{21}),
\end{aligned}
\tag{161}
$$

and intra-species interactions

$$
\begin{aligned}
L_j^\dagger L_j &= |a|^2(1 - e_j^{22})e_{j+1}^{22} + |b|^2e_j^{22}(1 - e_{j+1}^{22}) + |c|^2e_j^{22} - a^*ce_j^{12}e_{j+1}^{21} + c^*ae_j^{21}e_{j+1}^{12}, \\
\overline{L_j^\dagger L_j} &= |a|^2(1 - \widetilde{e}_j^{22})\widetilde{e}_{j+1}^{22} + |b|^2\widetilde{e}_j^{22}(1 - \widetilde{e}_{j+1}^{22}) + |c|^2\widetilde{e}_j^{22} - ac^*\widetilde{e}_j^{12}\widetilde{e}_{j+1}^{21} + ca^*\widetilde{e}_j^{21}\widetilde{e}_{j+1}^{12}.
\end{aligned}
\tag{162}
$$

In order to produce the cubic terms in $H_A^{(\epsilon)}$ we require

$$
a = -b, \quad ac^* = 1 - \epsilon\sin\theta, \quad ca^* = \sin\theta - 1.
\tag{163}
$$

Combining these with the requirement that $g = 0$ leads to

$$
\epsilon = \sin\theta = 1.
\tag{164}
$$

In this case the $A^{\pm}$ model reduces to free fermions. We have also investigated whether carrying out a similarity transformation $SH_A^{(\epsilon)}S^{-1}$ with

$$S = \prod_{j=1}^{L} \exp\left( \varphi e_j^{22}\widetilde{e}_j^{22} + j\left( \varphi_1 e_j^{11} + \varphi_2 e_j^{22} + \tilde{\varphi}_1 \widetilde{e}_j^{11} + \tilde{\varphi}_2 \widetilde{e}_j^{22} \right) \right) \tag{165}$$

may facilitate a Lindblad interpretation. The answer appears to be negative.

## 8 Discussion

In this work we have reported our findings for a search for Yang-Baxter integrable Lindblad equations. We have focused on translationally invariant situations where jump operators act on bonds or sites of a one dimensional chain. We have derived a superoperator representation for lattice models with both fermionic and bosonic degrees of freedom, and jump operators which can be bosonic or fermionic. In this representation the Lindblad equation takes the form of a imaginary time Schrödinger equation with a non-Hermitian "Hamiltonian" with local density, which can be thought of in terms of a two-leg ladder model of interacting spins or fermions. We have then investigated which Yang-Baxter integrable two-leg ladder models can be related to such Lindblad equations in a "direct" way. Our main result is that a wide class of generalized Hubbard models can be interpreted as Liouvillians of Lindblad equations. We traced this back to their integrability structure, which is based on gluing together certain solutions of the Yang-Baxter equation in a particular way. Some of the corresponding dissipative models are physically meaningful, an example being the infinite-U Hubbard model subject to on-site dephasing noise. As the jump operators in this class of models are Hermitian, the completely mixed state is a steady state in all cases. Using the Bethe Ansatz solution we have shown for a subclass of generalized Hubbard models that the Liouvillian gap vanishes like $L^{-2}$ as the thermodynamic limit is approached. The corresponding eigenstates correspond to particle-like "excitations" with quadratic dispersions, which suggests that the late-time behaviour in these models is likely to be diffusive.

We have identified a few Yang-Baxter integrable Lindblad equations that are not generalized Hubbard models by showing that certain known integrable Hamiltonians can be cast in the form of Liouvillians associated with a Lindblad equation. However, in most cases we have considered such mappings are not possible. As this is often difficult to see we have presented a non-trivial case of such a failure in the Alcaraz-Bariev two-parameter family of integrable models.

We stress that in this work we have focused on a particular "direct" relation between Liouvillians of Lindblad equations and Hamiltonians of Yang-Baxter integrable models. There are known cases where it is possible to establish such relationships by means of more complicated (non-local) maps [29]. Moreover, as we pointed out in section 3.3, one ought to allow for similarity transformations that maintain locality of the Hamiltonian density in integrable models when trying to establish relations with Lindblad equations. A systematic way of doing this is by considering invariances of the Yang-Baxter equation, *cf.* Chapter 12.2.5 of Ref. [41]. For example, given a solution $R(\lambda, \mu) \in \text{End}(\mathbb{C} \otimes \mathbb{C})$ of the Yang-Baxter equation other solutions can be obtained as

$$\left[ V(\mu) \otimes V(\lambda) \right] R(\lambda, \mu) \left[ V^{-1}(\lambda) \otimes V^{-1}(\mu) \right], \tag{166}$$

where $V(\lambda)$ is an invertible $n \times n$ matrix. This allows one to introduce additional free parameters in the resulting Hamiltonian. The latter will generally be non-Hermitian, but this is not a problem in the present context of Lindblad equations. It would be interesting to pursue

this line of enquiry further and a good starting point will be the models successfully related to Lindblad equations in this work.

In this we work we focused on identifying integrable Lindblad equations and only briefly explored using methods of quantum integrability to obtain physical properties. A good starting point for this is to determine the spectrum of the Liouvillian, which is given in terms of the solutions of the relevant Bethe Ansatz equations. It is well understood that the nature of solutions to Bethe Ansatz equations changes quite substantially when a parameter is made complex, as this results in the "scattering phases" acquiring magnitudes different from unity. In practice this means that the structure of solutions to the Bethe Ansatz equations, which is usually encoded in appropriate string hypotheses, must be revisited and typically becomes more involved. Even in the simplest case of the Hubbard model the structure of Bethe Ansatz roots for Liouvillian eigenstates with eigenvalues that have large real parts and non-zero imaginary parts appears to be non-trivial. We plan to report on this issue in a future publication. Ultimately one would like to determine the dynamics of general Green's functions

$$\text{Tr}\Big[\rho(t)E_{j_1}^{\alpha_1\beta_1}\dots E_{j_n}^{\alpha_n\beta_n}\Big] \tag{167}$$

for evolution from a given initial density matrix $\rho(0)$. In some of the cases discussed above this is relatively simple because the equations of motion for these Green's functions decouple and for two-point functions can thus either be integrated numerically or determined from the exact Liouvillian eigenstates in the two-particle sector [70]. In cases like the 3-state Maassarani model a more involved analysis is required and it would be interesting to investigate this case in more detail.

## Acknowledgements

We are grateful to F. Göhmann, H. Katsura and T. Prosen for very helpful discussions. This work was supported by the EPSRC under grant EP/S020527/1.

## A Structure of the Liouvillian for the most general jump operator acting on a bond

The most general two site bosonic jump operator with nearest-neighbour interactions is

$$L_j = \sum_{\alpha\beta}\Big(\lambda_{\alpha\beta}E_j^{\alpha\beta} + \lambda'_{\alpha\beta}E_{j+1}^{\alpha\beta}\Big) + \sum_{\alpha\beta\gamma\delta}\mu_{\alpha\beta\gamma\delta}E_j^{\alpha\beta}E_{j+1}^{\gamma\delta} \,. \tag{168}$$

This gives rise to interaction terms between the two legs of the ladder

$$L_j\overline{L_j^\dagger} = \mathcal{I}_j^{(2)} + \mathcal{I}_j^{(3)} + \mathcal{I}_j^{(4)} \,, \tag{169}$$

where $\mathcal{I}_j^{(n)}$ involves $n$ Hubbard operators $E_j^{\alpha\beta}$, $\widetilde{E}_j^{\alpha\beta}$. The interaction along a single rung of the ladder is

$$
\begin{aligned}
\mathcal{I}_j^{(2)} = \sum_{\substack{\alpha_1\beta_1\\\alpha_2\beta_2}}\Big(&\lambda_{\alpha_1\beta_1}\lambda_{\alpha_2\beta_2}^*E_j^{\alpha_1\beta_1}\widetilde{E}_j^{\alpha_2\beta_2} + \lambda_{\alpha_1\beta_1}\lambda_{\alpha_2\beta_2}^{'*}E_j^{\alpha_1\beta_1}\widetilde{E}_{j+1}^{\alpha_2\beta_2}\\
&+\lambda'_{\alpha_1\beta_1}\lambda_{\alpha_2\beta_2}^*E_{j+1}^{\alpha_1\beta_1}\widetilde{E}_j^{\alpha_2\beta_2} + \lambda'_{\alpha_1\beta_1}\lambda_{\alpha_2\beta_2}^{'*}E_{j+1}^{\alpha_1\beta_1}\widetilde{E}_{j+1}^{\alpha_2\beta_2}\Big) \,,
\end{aligned} \tag{170}
$$

while the three and four point interactions on a given plaquette are given by

$$
\begin{aligned}
\mathcal{I}_j^{(3)} &= \sum_{\substack{\alpha_1\beta_1\gamma_1\delta_1 \\ \alpha_2\beta_2}} \mu_{\alpha_1\beta_1\gamma_1\delta_1} E_j^{\alpha_1\beta_1} E_{j+1}^{\gamma_1\delta_1} \left( \lambda_{\alpha_2\beta_2}^* \widetilde{E}_j^{\alpha_2\beta_2} + \lambda_{\alpha_2\beta_2}^{'*} \widetilde{E}_{j+1}^{\alpha_2\beta_2} \right) \\
&\quad + \sum_{\substack{\alpha_1\beta_1 \\ \alpha_2\beta_2\gamma_2\delta_2}} \mu_{\alpha_2\beta_2\gamma_2\delta_2}^* \left( \lambda_{\alpha_1\beta_1} E_j^{\alpha_1\beta_1} + \lambda_{\alpha_1\beta_1}^{'} E_{j+1}^{\alpha_1\beta_1} \right) \widetilde{E}_j^{\alpha_2\beta_2} \widetilde{E}_{j+1}^{\gamma_2\delta_2} , \\
\mathcal{I}_j^{(4)} &= \sum_{\substack{\alpha_1\beta_1\gamma_1\delta_1 \\ \alpha_2\beta_2\gamma_2\delta_2}} \mu_{\alpha_1\beta_1\gamma_1\delta_1} \mu_{\alpha_2\beta_2\gamma_2\delta_2}^* E_j^{\alpha_1\beta_1} E_{j+1}^{\gamma_1\delta_1} \widetilde{E}_j^{\alpha_2\beta_2} \widetilde{E}_{j+1}^{\gamma_2\delta_2} .
\end{aligned} \tag{171}
$$

There are also interaction terms along the two legs of the ladder

$$
L_j^\dagger L_j = \sum_{\beta\gamma} \left[ \left( \sum_\alpha \lambda_{\alpha\beta} \lambda_{\alpha\gamma}^* \right) E_j^{\gamma\beta} + \left( \sum_\alpha \lambda_{\alpha\beta}^{'} \lambda_{\alpha\gamma}^{'*} \right) E_{j+1}^{\gamma\beta} \right] + \sum_{\alpha\beta\gamma\delta} \left[ f_{\alpha\beta\gamma\delta} E_j^{\alpha\beta} E_{j+1}^{\gamma\delta} + \text{h.c.} \right] ,
$$

$$
\overline{L_j^\dagger L_j} = \sum_{\beta\gamma} \left[ \left( \sum_\alpha \lambda_{\alpha\beta}^* \lambda_{\alpha\gamma} \right) \widetilde{E}_j^{\gamma\beta} + \left( \sum_\alpha \lambda_{\alpha\beta}^{'*} \lambda_{\alpha\gamma}^{'} \right) \widetilde{E}_{j+1}^{\gamma\beta} \right] + \sum_{\alpha\beta\gamma\delta} \left[ f_{\beta\alpha\delta\gamma} \widetilde{E}_{j+1}^{\gamma\delta} \widetilde{E}_j^{\alpha\beta} + \text{h.c.} \right] , \tag{172}
$$

where

$$
f_{\alpha\beta\gamma\delta} = \lambda_{\beta\alpha}^* \lambda_{\gamma\delta}^{'} + \sum_\eta \left[ \lambda_{\eta\alpha}^* \mu_{\eta\beta\gamma\delta} + \lambda_{\eta\gamma}^{'*} \mu_{\alpha\beta\eta\delta} \right] + \frac{1}{2} \sum_{\eta\nu} (-1)^{(\epsilon_\alpha+\epsilon_\beta)(\epsilon_\eta+\epsilon_\gamma)} \mu_{\nu\beta\eta\delta} \mu_{\nu\alpha\eta\gamma}^* . \tag{173}
$$

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
