# Peer review of "Yang-Baxter integrable Lindblad equations"

_SciPost Physics, doi:SciPost Phys. 8, 044 (2020)_

## Round 1 · Referee Report · Anonymous (Referee 1) · 2020-2-14

Strengths

see report

Weaknesses

see report: no obvious problems

Report

The authors present a very interesting study of Lindblad equations resulting in
integrable Liouvillians. The manuscript presents concrete results, path breaking
ideas, and open problems. I surely recommend the manuscript for publication in
SciPost.

The authors describe their approach as "direct". This means that they set up
Lindblad equations with or "without" a Hamiltonian, they add suitable jump
operators that result in a Liouvillian which in a natural manner can be
understood as a Hamiltonian on a doubled system (respectively a 2-leg ladder)
and "appear" to be known integrable Hamiltonians. Such an identification, of
course, may involve some similarity transformations.

The authors start by revisiting the appearance of the Hubbard model with
imaginary U parameter as Liouvillian. This is "obtained" by considering a
tight-binding model coupled to an environment by jump operators. A new result
is the use of a modified jump operator which leads to the Umeno, Shiroishi and
Wadati model.

In a similar manner, generalized Hubbard models like GL(N, M) Maassarani
models are used. A notable example is the 4-state Maassarani model. Also
GL(N^2) magnets are found to have associated Lindblad equations. An important
example of this is the GL(4) spin ladder.

Other examples of integrable quantum chains with associated Lindblad equations
are graded magnets. However, generalizations therof on the basis of
q-deformations seem to not qualify as integrable Liouvillians. And likewise
the Alcaraz-Bariev model "A" does not qualify. However, suitable similarity
transformations may change this understanding.

Some other open problem is posed by the continuum limit. The authors argue
that the only consistent scaling limit exists for vanishing interaction term
(a general feature independent of integrability).

The authors point out that n-particle Green’s functions fulfil simple, closed
evolution equations. Some of the constructed models, using Bethe Ansatz
techniques, show diffusive late-time dynamics.

I am convinced that the manuscript will serve as a starting point for many
future investigations of the presented models and for the search of new ones.

Requested changes

none
(some typos exist, but I think the authors will spot them)

A possibly incomplete list:

-- The customary approach in (-> is)
-- As we have mention (ed)
-- but must keep it finite in order to have describe (drop have?)

---

## Round 1 · Referee Report · Anonymous (Referee 2) · 2020-2-26

Strengths

see report

Weaknesses

A few minor comments (see report).

Report

In this paper, the authors present a systematic approach to constructing a variety of integrable Lindblad equations in the Yang-Baxter sense. This is achieved by first mapping Liouvillians associated with a Lindblad equation to non-Hermitian Hamiltonians on two-leg ladders and then identifying those Hamiltonians with previously known non-Hermitian integrable models. (The authors explain the strategy and procedure for this in a very pedagogical manner.)

Although the forms of the local Hamiltonians and the jump operators discussed are rather artificial at the cost of their integrability, some of them include physically meaningful ones such as the infinite-U Hubbard model with on-site dephasing noise. In addition, for the Maassarani models, the authors derive an analytical expression for the Liouvillian gap which scales with the system size $L$ as $1/L^2$. At the end of the paper, the authors list several classes of models for which their attempt to identify them with Liouvilians fails.

I think the paper addresses a currently important topic and paves a way towards a complete list of integrable Lindblad equations. I thus believe that the manuscript is suited for publication in SciPost, and suggest that the authors address a few issues below:

  • The title of Sec. 2.1: I suggest that the authors elaborate more on this decomposition in the main text rather than in the title. The authors might want to explain what $n_B$ and $n_F$ mean in the "bare" (non-super) models.

  • Scaling of the Liouvillian gap In Sec. 4.4.4 the authors discuss the finite-size scaling of the Liouvillian gap in the Maassarani model building on the string hypothesis. But since the set of Bethe ansatz equations (containing a pure-imaginary parameter) is pretty complicated, it is not quite obvious if the set of quantum numbers Eq. (99) really gives the lowest Liouvillian gap. Isn't it more appropriate to say that the analytic expression Eq. (101) just gives an upper bound on the Liouvilliang gap? Actually, the $1/L^3$ scaling has been observed for models with boundary dephasing. (See, e.g., Phys. Rev. E 92, 042143 (2015).)

  • Twisting the boundary conditions In Sec. 5.1.3, the authors identify a conjugated Liouvillian (Eq. (124)) with the integrable $GL(N^2)$ Hamiltonian with twisted boundary conditions and conclude the integrability of the Liouvillian. Although I do not doubt their conclusion, some readers might wonder whether twisting the boundary conditions will spoil the integrability. The authors might want to add a few words on it in the revision.

  • Continuum limit Sec. 6 addresses the issue about the continuum limit of the Liouvillian. I guess $c^\dagger_j$ and $c_j$ in Eq. (144) refer to the creation and annihilation operators of a fermion at site $j$. But this seems not defined. Another problem is: I do not quite see the point of this section. I think the continuum limit Eq. (149) is valid only at the zero-density limit in which the divergence of the coefficient in the first term would not be a problem. Also, I wonder what if the authors linearize the dispersion around the Fermi points and then take the continuum limit, which is standard in the Luttinger-liquid/bosonization approach. Does this situation make the scaling limit nontrivial? What's wrong with the previous field-theory approach discussed, e.g., in a review, Rep. Prog. Phys. 79, 096001 (2016)? (See Sec. V.)

Requested changes

As another referee also says there are typos.

  • Maassarani introduced a class of integrable 2n-state models -> Maassarani introduced a class of integrable $2n$-state models

  • We have focussed on translationally invariant ... -> We have focused on translationally invariant ...

  • validity: high
  • significance: good
  • originality: high
  • clarity: high
  • formatting: excellent
  • grammar: excellent

Author:  Fabian Essler  on 2020-03-08  [id 754]

(in reply to Report 2 on 2020-02-26)

We thank the referee for their report and constructive comments. The referee raised four issues to which we now reply in turn.

(1) We have followed the referee's suggestion to add explanations on section 2.1.

(2) Scaling of the Liouvillian gap. We made no claims that the states we constructed have the largest non-zero eigenvalues, we merely stated that the Liouvillian gap vanishes in the thermodynamic limit. The latter indeed follows from our construction of particular eigenstates with a gap the scales as $L^{-2}$. In order to avoid misunderstandings we have added a sentence stating that there well could be states with smaller Liouvillian gaps.

(3) Twisting the boundary conditions. It is well known that twisting the boundary conditions for the $GL(N^2)$ models is compatible with integrability. We have added some references on different ways (quantum inverse scattering method, co-ordinate Bethe Ansatz) of showing this.

(4) Continuum limit. The purpose of this section is to point out that the standard way of deriving integrable quantum field theories from integrable lattice models by taking appropriate scaling limits does not work as we are dealing with a master equation rather than a Schr\"odinger equation. We have added a sentence to make this more clear.

We disagree with the referee's statement that "the continuum limit Eq. (149) is valid only at the zero-density limit in which the divergence of the coefficient in the first term would not be a problem." The offending operator is the particle number, not the particle density. Hence adding even a single particle would generate a divergent contribution once we take $\gamma\to\infty$. This can of course also be seen directly from the Bethe Ansatz solution of the
model. Linearizing around the Fermi points at finite density leads to the same problem. The reference quoted by the referee, Rep. Prog. Phys. 79, 096001 (2016), considers a model of interacting bosons rather than fermions, and applies bosonisation techniques without taking a scaling limit as we do here, i.e. it keeps $\gamma$ finite. As we understand it the aim of this approach is to obtain an approximate description of the original lattice model, applicable for an appropriate class of initial density matrices and sufficiently short time scales. For our purposes the problem with not taking a scaling limit is that all higher derivative terms a priori have to be retained in the Liouvillian and they break integrability. We have changed "continnum limts" to "scaling limits" in the title of the section in order to stress that we are interested in the latter rather than the former.

Attachment:

reply.pdf

---

## Round 2 · Referee Report · Anonymous (Referee 2) · 2020-3-9

Strengths
see report
Weaknesses
a minor comment (see report)
Report
I have examined the revised manuscript and the authors' replies to the comments raised by the referees. I think their responses are convincing/pertinent and the paper has improved the quality/readability.
My only concern is their reply to my comment (2). As far as I understand, the references they cite ([41, 53, 54]) are concerned with simpler spin-1/2 Heisenberg-Ising and Hubbard models rather than a more general $GL(N^2$) model. As I said in my previous report, I do not doubt the conclusion. But I just wonder if the authors could give more appropriate references for higher-rank models so that the reader can refer to the details.
My only concern is their reply to my comment (2). As far as I understand, the references they cite ([41, 53, 54]) are concerned with simpler spin-1/2 Heisenberg-Ising and Hubbard models rather than a more general $GL(N^2$) model. As I said in my previous report, I do not doubt the conclusion. But I just wonder if the authors could give more appropriate references for higher-rank models so that the reader can refer to the details.
Requested changes
page 14: the n-state Maassarani models
-> the $n$-state Maassarani models

---

## Round 2 · Author Response

We thank the referees for their reports and constructive comments. The second referee raised four issues to which we now reply in turn.
(1) We have followed the referee's suggestion to add explanations on section 2.1.
(2) Scaling of the Liouvillian gap. We made no claims that the states we constructed have the largest non-zero eigenvalues, we merely stated that the Liouvillian gap vanishes in the thermodynamic limit. The latter indeed follows from our construction of particular eigenstates with a gap the scales as $L^{-2}$. In order to avoid misunderstandings we have added a sentence stating that there well could be states with smaller Liouvillian gaps.
(3) Twisting the boundary conditions. It is well known that twisting the boundary conditions for the $GL(N^2)$ models is compatible with integrability. We have added some references on different ways (quantum inverse scattering method, co-ordinate Bethe Ansatz) of showing this.
(4) Continuum limit. The purpose of this section is to point out that the standard way of deriving integrable quantum field theories from integrable lattice models by taking appropriate scaling limits does not work as we are dealing with a master equation rather than a Schr\"odinger equation. We have added a sentence to make this more clear.
We disagree with the referee's statement that "the continuum limit Eq. (149) is valid only at the zero-density limit in which the
divergence of the coefficient in the first term would not be a problem." The offending operator is the particle number, not the
particle density. Hence adding even a single particle would generate a divergent contribution once we take $\gamma\to\infty$. This can of course also be seen directly from the Bethe Ansatz solution of the model. Linearizing around the Fermi points at finite density leads to the same problem. The reference quoted by the referee, Rep. Prog. Phys. 79, 096001 (2016), considers a model of interacting bosons rather than fermions, and applies bosonisation techniques without taking a scaling limit as we do here, i.e. it keeps $\gamma$ finite. As we understand it the aim of this approach is to obtain an approximate description of the original lattice model, applicable for an appropriate class of initial density matrices and sufficiently short time scales. For our purposes the problem with not taking a scaling limit is that all higher derivative terms a priori have to be retained in the Liouvillian and they break integrability. We have changed "continnum limts" to "scaling limits" in the title of the section in order to stress that we are interested in the latter rather than the former.
(1) We have followed the referee's suggestion to add explanations on section 2.1.
(2) Scaling of the Liouvillian gap. We made no claims that the states we constructed have the largest non-zero eigenvalues, we merely stated that the Liouvillian gap vanishes in the thermodynamic limit. The latter indeed follows from our construction of particular eigenstates with a gap the scales as $L^{-2}$. In order to avoid misunderstandings we have added a sentence stating that there well could be states with smaller Liouvillian gaps.
(3) Twisting the boundary conditions. It is well known that twisting the boundary conditions for the $GL(N^2)$ models is compatible with integrability. We have added some references on different ways (quantum inverse scattering method, co-ordinate Bethe Ansatz) of showing this.
(4) Continuum limit. The purpose of this section is to point out that the standard way of deriving integrable quantum field theories from integrable lattice models by taking appropriate scaling limits does not work as we are dealing with a master equation rather than a Schr\"odinger equation. We have added a sentence to make this more clear.
We disagree with the referee's statement that "the continuum limit Eq. (149) is valid only at the zero-density limit in which the
divergence of the coefficient in the first term would not be a problem." The offending operator is the particle number, not the
particle density. Hence adding even a single particle would generate a divergent contribution once we take $\gamma\to\infty$. This can of course also be seen directly from the Bethe Ansatz solution of the model. Linearizing around the Fermi points at finite density leads to the same problem. The reference quoted by the referee, Rep. Prog. Phys. 79, 096001 (2016), considers a model of interacting bosons rather than fermions, and applies bosonisation techniques without taking a scaling limit as we do here, i.e. it keeps $\gamma$ finite. As we understand it the aim of this approach is to obtain an approximate description of the original lattice model, applicable for an appropriate class of initial density matrices and sufficiently short time scales. For our purposes the problem with not taking a scaling limit is that all higher derivative terms a priori have to be retained in the Liouvillian and they break integrability. We have changed "continnum limts" to "scaling limits" in the title of the section in order to stress that we are interested in the latter rather than the former.

---

## Round 3 · Referee Report · Anonymous · 2020-3-10

Strengths
see report
Weaknesses
no problems
Report
Concerning the reservation in my previous report, the authors have added appropriate references addressing the higher-rank Heisenberg model with toroidal boundary conditions. I think the manuscript is now suited for publication in SciPost.

---

## Round 3 · List of Changes

Reference added

---

## Editorial Decision

published